## RESEARCH ARTICLE

# What impact do new homologs have on detecting interdomain horizontal gene transfer in eukaryotes? A reassessment of Katz (2015)

Kevin Aguirre-Carvajal[1,2] and Vinicio Armijos-Jaramillo[2,3,*]

## ABSTRACT

The role of interdomain horizontal gene transfer (iHGT) in eukaryotic genome evolution remains a subject of ongoing debate. Numerous studies have reported prokaryote-to-eukaryote transfer events, yet the extent to which these inferences are sensitive to taxon sampling and methodological choices remains unclear. In this study, we performed an independent phylogenetic analysis of the 1138 candidate genes previously proposed by Katz (2015), using updated homology searches, expanded taxon sampling, and different iHGT detection pipelines. Under the interpretative framework applied here, approximately 30% of candidates exhibited phylogenetic support for iHGT. The remaining candidates were classified as inconclusive, as their phylogenetic patterns were broader or ambiguous and compatible with alternative evolutionary scenarios, including cyanobacterial affinity consistent with endosymbiotic gene transfer, differential gene loss, incomplete lineage sorting, absent or limited donor representation. In many cases, the recovery of homologs from additional eukaryotic major clades transformed apparently lineage-restricted genes into multi-clade distributions, illustrating the strong influence of taxon sampling on iHGT inference. These findings underscore the sensitivity of horizontal gene transfer detection to database completeness, analytical thresholds, and evolutionary context. Rather than providing a definitive count of transfer events, this study highlights how expanding genomic resources and methodological choices shape interpretations of interdomain gene transfer in eukaryotes.

KEY WORDS: Interdomain horizontal gene transfer, Eukaryotes, Genome evolution, Phylogeny, New homologs

## INTRODUCTION

Horizontal gene transfer (HGT) is a well-established mechanism for generating genetic variability in bacteria (Brito, 2021). Most documented cases involve gene transfers between bacterial species, whereas reports of HGT from eukaryotes to bacteria are far less common. This scarcity is expected, given the substantial genetic incompatibilities between such evolutionary distinct groups. Interestingly, despite these incompatibilities, interdomain HGT (iHGT) from bacteria to eukaryotes is frequently reported in the scientific literature. This is surprising, as eukaryotic genomes possess multiple barriers that could hinder iHGT, including differences in gene promoter recognition, intron processing, codon usage bias, and the presence of a nucleus that encapsulates genetic material (Fitzpatrick, 2011). In fungi specifically, the acquisition of foreign genetic material may trigger meiotic silencing by unpaired DNA, further complicating the successful integration of horizontally transferred genes (Hammond, 2017).

The recent surge in reported cases of iHGT in the scientific literature (Katz, 2015; Kwak et al., 2023; Liu et al., 2025; Moran et al., 2012) may create the impression that this phenomenon is widespread in eukaryotic evolution. For instance, Cote-L'Heureux et al. (2022) identified 306 iHGT events in eukaryotes, while Kwak et al. (2023) reported at least seven bacterial-derived genes in the genome of the psyllid *Bactericera cockerelli*. Lai et al. (2023) found 161 orthogroups acquired by nematodes from various kingdoms, particularly bacteria. Liu et al. (2025) analyzed 750 fungal genes and identified 20,093 prokaryotic-derived genes. Additionally, Katz (2015) conducted a comprehensive study on the presence of iHGT and endosymbiotic gene transfer (EGT) across major eukaryotic clades, reporting 1138 candidate genes potentially transferred from bacteria to eukaryotes. This analysis was based on 487 eukaryotic genomes, along with 303 bacterial and 118 archaeal genomes. Given this extensive dataset, it appears that iHGT has played a role in eukaryotic evolution. However, the actual impact of this process remains a topic of ongoing debate (Aguirre-Carvajal et al., 2024; Ku and Martin, 2016; Martin, 2017). Aguirre-Carvajal et al. (2024) suggest that as genomic databases expand, the number of iHGT cases reported tends to decline, highlighting the need for a continued reassessment of this phenomenon.

To evaluate this premise, we examined the 1138 iHGT events reported by Katz (2015) using updated homology searches, expanded taxon sampling, and revised phylogenetic evaluation criteria. Rather than replicating the original analytical pipeline, our approach applies an alternative and more explicit set of phylogenetic criteria to identify gene candidates with robust support for prokaryote-to-eukaryote transfer. Our goal was to assess how expanded genomic resources and methodological differences influence the inferred prevalence of iHGT.

## RESULTS

### Effects of analytical framework on iHGT candidate inference

Katz (2015) reported 1138 iHGT events from prokaryotes into eukaryotes, classified according to their presence in either a single major eukaryotic clade (1MC) or multiple major clades (2 and 3MCs). In the present study, all reported candidates were systematically

[1]Department of Computer Science and Information Technologies, Faculty of Computer Science, CITIC Research Center of Information and Communication Technologies, University of A Coruña, Campus Elviña s/n, 15071 A Coruña, Spain. [2]Bio-Cheminformatics Research Group, Universidad de Las Américas, Quito 170513, Ecuador. [3]Carrera de Ingeniería en Biotecnología, Facultad de Ingeniería y Ciencias Aplicadas, Universidad de Las Américas, Quito 170513, Ecuador.

*Author for correspondence (vinicio.armijos@udla.edu.ec)

K.A.-C., 0009-0004-5143-6856; V.A.-J., 0000-0003-2965-2515

analyzed to assess their phylogenetic support and clade distribution using different iHGT detection strategies.

In this study, organisms were classified into MCs and associated minor clades (mcs), as defined in Table 1.

According to the results obtained from the AVP tool (see Materials and Methods), the 1138 iHGT candidates were classified into three categories based on their phylogenetic signal. A total of 466 candidates were classified as iHGT, indicating that the topology of the inferred phylogenetic trees was consistent with a HGT event. In contrast, 661 candidates were classified as No HGT, reflecting the absence of a detectable iHGT signal in their phylogenetic reconstructions. The remaining 11 candidates were categorized as Complex, corresponding to cases in which the AVP tool was unable to conclusively determine whether the observed tree topology supported an iHGT event. The output of the AVP tool is summarized in Table S1.

Among the 466 candidates classified as iHGT by the AVP analysis, 70% had originally been reported by Katz as involving 1MC, 29% as involving 2MCs, and 1% as involving 3MCs. In contrast, among the 661 candidates classified as No HGT, 53% were originally assigned to 1MC, 38% to 2MCs and 9% to 3MCs. Finally, of the 11 candidates categorized as Complex, 45% were originally classified as 1MC and 55% as 2MCs. These distributions indicate that candidates originally assigned to higher MC categories were more frequently classified as No or Complex in the AVP analysis.

Based on manual phylogenetic inspection, iHGT candidates were classified into four major categories, as summarized in Table 2. Of the 1138 candidates analyzed, 344 presented clear phylogenetic support for iHGT. A total of 94 were identified as putative EGT (endosymbiotic gene transfer), as their phylogenetic trees showed clustering with Cyanobacteria and no other bacterial group. In addition, 683 candidates exhibited inconclusive phylogenetic patterns, while 17 were flagged as potential contamination. The complete classification for all candidates is provided in Table S2.

**Table 1. Classification scheme of MCs and associated mcs applied in this study**

| MC | mc | Code (MC_mc) |
|---|---|---|
| Opisthokonta | Aphelida | Op_ap |
| Opisthokonta | Choanoflagellata | Op_ch |
| Opisthokonta | Filasterea | Op_fi |
| Opisthokonta | Fungi | Op_fu |
| Opisthokonta | Ichthyosporea | Op_ic |
| Opisthokonta | Metazoa | Op_me |
| Opisthokonta | Rotosphaerida | Op_ro |
| Sar | Alveolata | Sr_al |
| Sar | Rhizaria | Sr_rh |
| Sar | Stramenopiles | Sr_st |
| Amoebozoa | Discosea | Am_di |
| Amoebozoa | Evosea | Am_ev |
| Amoebozoa | Amoebozoa incertae sedis | Am_is |
| Archaeplastida | Viridiplantae | Pl_gr |
| Archaeplastida | Rhodophyta | Pl_rh |
| Archaeplastida | Glaucocystophyceae | Pl_gl |
| Excavata | Metamonada | Ex_me |
| Excavata | Discoba | Ex_di |
| Unclassified Eukaryota | Apusozoa | UE_ap |
| Unclassified Eukaryota | Cryptophyceae | UE_cr |
| Unclassified Eukaryota | Haptista | UE_ha |
| Prokaryota | Bacteria | Pr_Ba |
| Prokaryota | Archaea | Pr_Ar |

MC and mc classifications were derived from the NCBI Taxonomy Browser (accessed 10 December 2025; https://www.ncbi.nlm.nih.gov/Taxonomy/Browser/wwwtax.cgi).

**Table 2. Classification categories resulting from the manual assessment of iHGT candidates**

| Category | Phylogenetic pattern | Number |
|---|---|---|
| iHGT | iHGT | 344 |
| Putative EGT | Cyanobacterial homology | 94 |
| Inconclusive | Limited donor evidence | 178 |
|  | No prokaryotic homologs | 41 |
|  | Multiple major clades | 214 |
|  | Patchy phylogeny | 204 |
|  | No eukaryotic homologs | 46 |
| Potential contamination | Potential contamination | 17 |

The 344 candidates with iHGT pattern shared a common phylogenetic signature defined during manual inspection. Specifically, candidates were considered as iHGT when their phylogenetic trees showed a eukaryotic clade restricted to a single major lineage that was nested within, or placed as sister to, a prokaryotic clade, with no additional homologs detected in other eukaryotic MCs. In these cases, the overall topology was compatible with a single acquisition from prokaryotes into the focal eukaryotic lineage.

The 94 candidates exhibiting exclusive homology to Cyanobacteria shared a distinct phylogenetic signature. In these cases, the eukaryotic sequences formed a clade that was either nested within or sister to cyanobacterial homologs, and no other bacterial groups were recovered as close relatives. Such a topology is consistent with a cyanobacterial origin of the gene and is compatible with putative EGT from the plastid ancestor.

Because plastids derive from a cyanobacterial endosymbiont, genes acquired during plastid establishment are expected to display precisely this phylogenetic pattern. However, this topology alone does not allow discrimination between gene transfer associated with the ancestral plastid endosymbiosis and transfers from cyanobacterial lineages.

While this topology is strongly suggestive of putative EGT, it does not, on its own, establish the exact mechanistic route of transfer; rather, it indicates a specific phylogenetic relationship with Cyanobacteria distinct from broader prokaryotic signals observed in other candidates. Accordingly, these cases were interpreted conservatively as compatible with putative EGT, while acknowledging that cyanobacterial transfer scenario cannot be formally excluded based on single-gene phylogenies. Within the Inconclusive category, five distinct phylogenetic patterns were identified. A subset of 178 candidates exhibited a limited number of prokaryotic homologs, resulting in phylogenetic topologies in which the inferred direction of transfer appeared more consistent with a eukaryote-to-prokaryote scenario rather than the prokaryote-to-eukaryote transfer. In addition, 41 candidates lacked detectable prokaryotic homologs altogether, indicating an absence of evidence supporting a prokaryotic origin for these genes.

Respectively, both patterns were classified as 'Limited donor evidence' and 'No prokaryotic homologs'. Since the scarcity of prokaryotic sequences reduces phylogenetic resolution and weakens confidence in inferring the directionality of transfer, these cases do not provide sufficient evidence to robustly support either scenario. Therefore, they were classified as inconclusive rather than being interpreted as definitive iHGT or rejected outright.

Another group of 214 candidates displayed phylogenetic trees in which the query sequence formed a monophyletic group with sequences belonging to multiple eukaryotic MCs, rather than being restricted to a single clade. The observed topology is more parsimoniously explained by vertical inheritance followed by

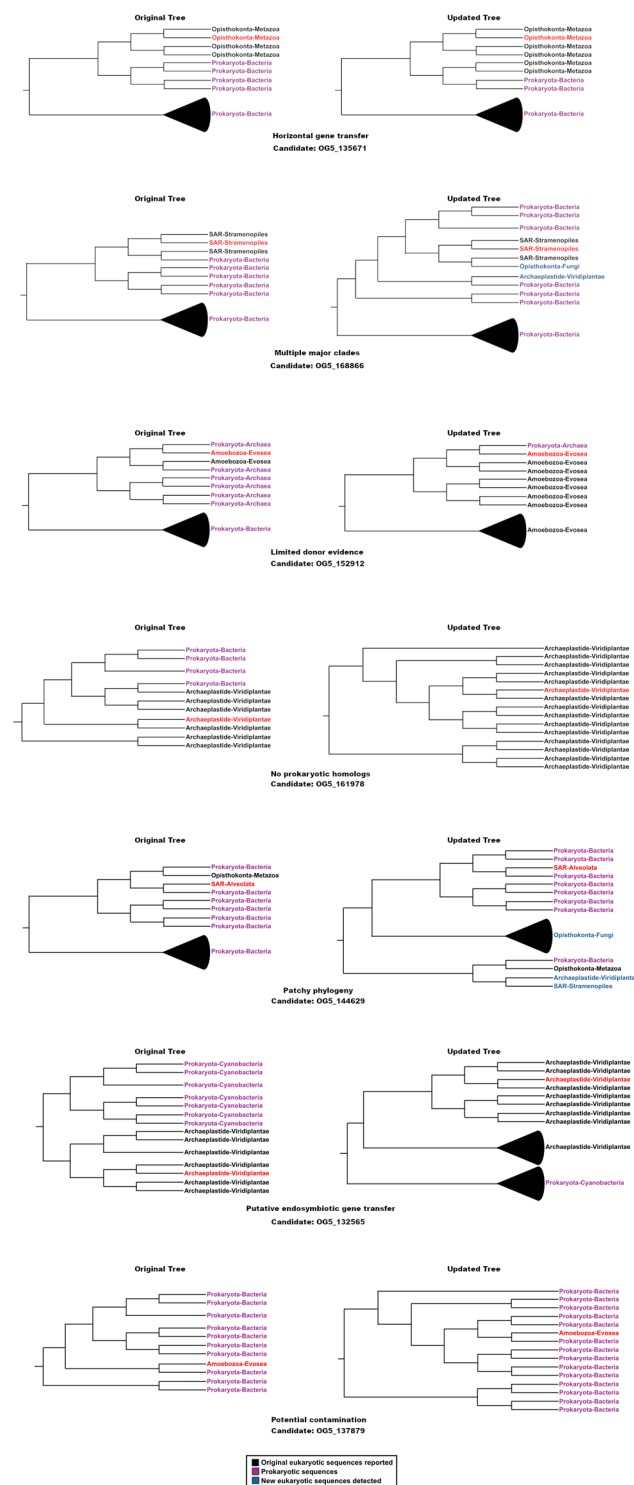

**Fig. 1. Schematic overview of seven iHGT candidates evaluated in this study.** For each candidate, the phylogenetic tree originally reported by Katz (left) is shown alongside the corresponding updated tree reconstructed in this study (right). Each candidate (e.g. OG5_XXXXX) represents an example of a broader phylogenetic pattern identified, including supported interdomain HGT (iHGT), presence of multiple major eukaryotic clades, limited donor evidence, absence of prokaryotic homologs, patchy phylogeny, putative EGT and potential contamination. Colors indicate taxonomic origin and sequence status: prokaryotic sequences are shown in purple, original eukaryotic sequences reported by Katz in black, newly detected eukaryotic homologs in blue, and the focal eukaryotic candidate sequence in red.

differential gene loss across lineages. Whereas an iHGT hypothesis would require invoking an ancient transfer from bacteria into an early eukaryotic ancestor accompanied by widespread secondary losses. Given that these candidates are compatible with alternative evolutionary scenarios, they were classified as 'Multiple MC' and excluded from the iHGT category.

A further group of 204 candidates were assigned to the 'Patchy phylogeny' pattern when the query sequence failed to form a monophyletic group with sequences from other MCs and instead showed homologs distributed across different positions in the tree. Also, such phylogenetic patterns can be explained by several evolutionary scenarios, including differential gene loss, incomplete lineage sorting, or iHGT.

Because single-gene phylogenies do not permit a decisive distinction among competing evolutionary scenarios, and the observed topologies do not provide support for prokaryote-to-eukaryote transfer, candidates exhibiting 'Multiple MC' or 'Patchy phylogeny' patterns were classified as Inconclusive rather than as supported iHGT events.

Additionally, 46 candidates lacked an identifiable eukaryotic query sequence, based on the IDs provided by Katz. Because the identification of iHGT requires a demonstrable eukaryotic gene embedded within a predominantly prokaryotic phylogenetic context, the absence of a detectable eukaryotic query precluded formal classification of these cases as iHGT. Accordingly, they were classified as inconclusive under the category 'No eukaryotic homologs'.

Finally, 17 candidates were classified as 'Potential contamination' category. These candidates were characterized by genomic contexts in which the flanking genes in the contig where the candidate is located exhibited strong homology to prokaryotic lineages, indicating that the observed signal most likely reflects contamination rather than bona fide HGT. After accounting for these cases, approximately 30% presented phylogenetic support for prokaryote-to-eukaryote gene transfer under the criteria applied here. The phylogenetic trees and multiple sequence alignments underlying these classifications are available in the Zenodo repository (https://doi.org/10.5281/zenodo.18435322). These patterns are schematically illustrated in Fig. 1.

Upon completing the phylogenetic classification using both AVP and our phylogenetic pipeline, we observed marked method-dependent differences in iHGT support. Compared with AVP, our pipeline detected 122 fewer candidates, representing a 26% reduction in iHGT calls. Agreement between methods was partial: 48% of the iHGT candidates supported by our pipeline were also detected by the automated AVP pipeline.

## Impact of database expansion on iHGT detection

The classification of a large fraction of candidates as 'Multiple MC' or 'Patchy phylogeny' suggests that the expanded detection of homologs across eukaryotic lineages fundamentally alters the apparent phylogenetic signal underlying many previously proposed iHGT events. In particular, the recovery of additional homologs genes from multiple eukaryotic clades weakens support for a lineage-restricted acquisition and instead favors explanations based on ancestral presence and differential loss. To test whether increased taxon sampling contributes to this shift in phylogenetic interpretation, we evaluated the relationship between changes in taxon representation between databases and the number of candidates detected per taxonomic group. This was assessed using correlation analyses and logistic regression to identify which taxonomic groups most strongly influence candidate classification.

Fig. 2 illustrates the relationship between changes in taxonomic sampling and changes in the number of inferred iHGT candidates

Biology Open

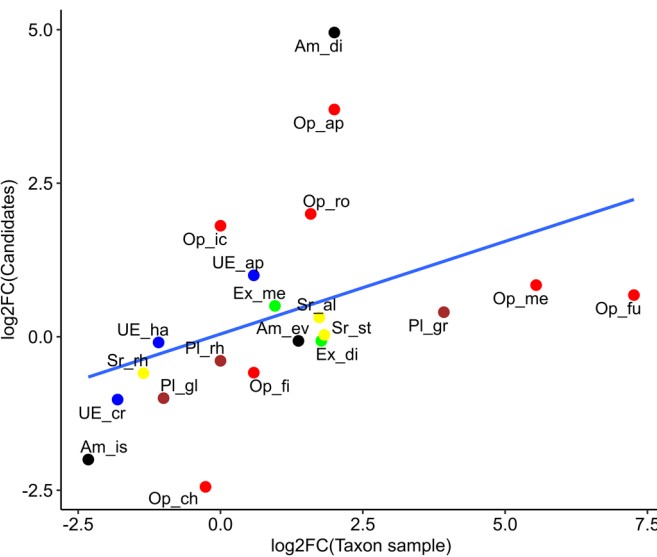

**Fig. 2. Relationship between the fold change in taxon sampling and the fold change in the number of candidate genes across taxonomic groups.** Each point represents a taxonomic group, with colors indicating major eukaryotic lineages: Opisthokonta (red), Unclassified Eukaryota (blue), SAR (yellow), Archaeplastida (brown), Amoebozoa (black), and Excavata (green). Fold changes are shown on a log$_2$ scale. Data used to generate this figure are provided in Table S4.

**Table 3. Effect of taxonomic group presence on the probability of detecting additional MCs.**

| MC_mc | P_adjust | prob1 | prob2 |
|---|---|---|---|
| Op_fu | 4.02×10$^{-56}$ | 0.09 | 0.59 |
| Sr_al | 1.37×10$^{-50}$ | 0.21 | 0.83 |
| Sr_st | 1.68Ex10$^{-44}$ | 0.22 | 0.76 |
| Op_me | 6.55×10$^{-44}$ | 0.17 | 0.60 |
| Pl_gr | 6.41×10$^{-22}$ | 0.19 | 0.47 |
| Pl_rh | 1.11×10$^{-15}$ | 0.29 | 0.78 |
| Am_ev | 4.67×10$^{-15}$ | 0.29 | 0.74 |
| Ex_me | 3.74×10$^{-12}$ | 0.30 | 0.91 |
| Ex_di | 6.34×10$^{-11}$ | 0.30 | 0.82 |
| UE_ha | 7.81×10$^{-08}$ | 0.31 | 0.84 |
| Sr_rh | 5.30×10$^{-07}$ | 0.31 | 0.74 |
| UE_cr | 3.24×10$^{-05}$ | 0.32 | 0.83 |
| Am_di | 0.000942 | 0.31 | 0.97 |
| Op_ch | 0.541205 | 0.33 | 0.75 |
| Op_ic | 0.689939 | 0.33 | 0.83 |
| Op_ap | 1 | 0.33 | 1.00 |
| Op_fi | 1 | 0.33 | 1.00 |
| Op_ro | 1 | 0.33 | 1.00 |
| Pl_gl | 1 | 0.33 | 0.58 |
| UE_ap | 1 | 0.32 | 1.00 |

For each taxonomic group (MC_mc), an independent logistic regression model was fitted using taxon presence/absence as a binary predictor. *P_adjust* indicates the adjusted *P*-value associated with the predictor coefficient, accounting for multiple testing. *prob1* represents the predicted probability of detecting an increase in MCs when the taxonomic group is absent, while *prob2* represents the predicted probability when the agroup is present. Probabilities were obtained by applying the inverse-logit transformation to the corresponding model coefficients.

across mcs. Overall, increased taxon sampling is associated with a higher proportion of candidates not meeting our criteria for iHGT classification, suggesting that differences in detected candidate distributions among clades are partly driven by database coverage. This pattern is supported by a significant Spearman rank correlation between log$_2$ fold changes in taxon sampling and candidate counts (ρ=0.71, *P*=0.0002), revealing a strong monotonic association between the two variables.

Importantly, both the scale and direction of these changes differ markedly among clades. Lineages that underwent substantial increases in taxon sampling frequently exhibited pronounced declines in lineage-restricted candidates, often accompanied by the reassignment of previously single-clade candidates into broader, multi-clade distributions. In contrast, clades with limited changes in sampling displayed comparatively minor shifts. Together, these results demonstrate that database expansion strongly influences iHGT inference and frequently weakens support for candidates previously identified under more limited taxonomic sampling.

To assess whether the presence of specific taxonomic groups is associated with changes in the number of inferred MCs, we fitted independent logistic regression models for each group. In each model, the presence or absence of a given taxon was used as a binary predictor, and the response variable indicated whether a candidate exhibited an increase in the number of MCs under the updated dataset. Detailed presence–absence data for all taxonomic groups are provided in Table S3.

The analysis revealed that 13 taxonomic groups had a statistically significant effect (*P*<0.05) on the probability of observing an increase in MCs (Table 3). In contrast, seven groups showed no significant association with increases in MC detection. These non-significant groups included five Opisthokonta lineages (Choanoflagellata, Ichthyosporea, Filasterea, Aphelida, and Rotosphaerida), one Archaeplastida lineage (Glaucocystophyceae), and one unclassified eukaryotic group (Apusozoa).

For taxonomic groups with significant effects, the presence of the group was consistently associated with a marked increase in the predicted probability of MC expansion. For example, in the case of fungi (Opisthokonta), the probability of detecting additional MCs increased from 0.09 in their absence to 0.59 when fungal homologs were present (*P*<0.05). Similar patterns were observed across Amoebozoa (Discosea and Evosea), SAR (Stramenopiles, Alveolata and Rhizaria), Excavata (Metamonada and Discoba), Archaeplastida (Rhodophyta) and Unclassified eukaryote (Haptista and Cryptophyceae) lineages, where predicted probabilities exceeded 0.74 when the focal taxon was present. Viridiplantae (Archaeplastide) and Metazoa (Opisthokonta) exhibited more moderate effects, with predicted probabilities of 0.47 and 0.60, respectively (Table 3).

Overall, these results indicate that the inclusion of specific eukaryotic groups – particularly those with expanded genomic representation – substantially increases the likelihood of detecting additional major clades among iHGT candidates, highlighting the strong influence of taxon sampling on phylogenetic interpretation.

The results from AVP and the manual phylogenetic assessment show that a subset of candidates presented evidence of iHGT support under the updated analytical framework. Approximately 40% and 30% of the candidates were classified as iHGT based on AVP and manual phylogenetic assessment, respectively. This reduction can be explained by several phylogenetic patterns; however, the most influential factor appears to be the emergence of homologs from additional eukaryotic MCs. This pattern is likely driven by differences in taxon sampling between databases and by the increased availability of genomic data at the time of the present analysis. Notably, comparison between AVP and the manual pipeline revealed limited concordance in iHGT support, highlighting the sensitivity of iHGT inference to the analytical framework employed.

## DISCUSSION

In this study, all 1138 iHGT candidates reported by Katz (2015) were systematically analyzed using different methods, updated homology searches, expanded taxon sampling, and revised phylogenetic evaluation criteria. This assessment provides an updated perspective on the phylogenetic patterns associated with prokaryote-to-eukaryote gene transfer.

Our analysis indicates that, under the criteria applied here, only a subset of the previously proposed candidates exhibits a clear phylogenetic pattern consistent with iHGT. Based on manual phylogenetic inspection, approximately 30% of candidates showed evidence to support iHGT pattern. The remaining cases displayed alternative or ambiguous phylogenetic patterns, including putative EGT, limited donor representation, absence of prokaryotic or eukaryotic homologs, patchy phylogenetic distributions, or potential contamination. These results are broadly consistent with the conclusions of Aguirre-Carvajal et al. (2025, 2024) and the ideas proposed by Martin (2017), and they underscore the importance of iterative evaluation as genomic resources expand.

Furthermore, the observed limited occurrence of lateral gene transfer in eukaryotes is consistent with known biological barriers that prevent the integration of foreign genetic material (Fitzpatrick, 2011). Despite this, numerous studies have reported iHGT cases in the scientific literature (Cote-L'Heureux et al., 2022; Katz, 2015; Kwak et al., 2023; Lai et al., 2023; Liu et al., 2025; Marcet-Houben and Gabaldón, 2010; Moran et al., 2012). Huang (2013) explored in detail the mechanisms by which these barriers might be bypassed, potentially explaining the high number of reported iHGT events.

Given this apparent contradiction, iHGT in eukaryotes remains a highly debated topic (Boto, 2018; Bremer et al., 2025; Ku et al., 2015; Ku and Martin, 2016; Leger et al., 2018; Martin, 2017; 2018; Roger, 2018). However, its impact on eukaryotic genomes remains an important and unresolved question that warrants further investigation. Katz (2015) made a significant effort to quantify iHGT in eukaryotes, yet with the continuous expansion of genomic data, all reported prokaryote-to-eukaryote iHGT cases require updated assessment.

Notably, Katz (2015) applied this same rationale when reevaluating the iHGT cases reported by Becker et al. (2008), which described gene transfers from Chlamydiae to Archaeplastida. Upon reanalysis, only two of the original 38 cases were validated. Similarly, Katz identified new homologs in different major clades for a previously claimed synapomorphy in Opisthokonta (tyrosyl-tRNA synthetase) (Huang et al., 2005). Our study observes the same pattern in the reassessment of Katz's iHGT candidates, reinforcing the notion that iHGT cases must be continuously evaluated as database sizes increase.

A central finding of this study is the strong influence of taxon sampling on inferred MC distributions. More than one-third of the candidates exhibited an increase in the number of detected eukaryotic MCs relative to Katz's original report, frequently involving lineages that were underrepresented at the time of the original analysis. This effect is particularly pronounced in Opisthokonta (Fungi and Metazoa) and Archaeplastida (Viridiplantae), for which Katz (2015) relied on taxon sample sizes of 40, 61, and 61 species, respectively, whereas the present analysis incorporates 6295 fungal, 2894 metazoan, and 940 Viridiplantae proteomes. Logistic regression analyses further demonstrate that the inclusion of specific taxonomic groups – especially those that have undergone substantial recent expansion in genomic representation – significantly increases the probability of detecting additional MCs. Together, these results indicate that the lack of certain eukaryotic homologs is often a consequence of incomplete sampling rather than definitive evidence for recent horizontal acquisition.

Katz (2015) concluded that most iHGT events into eukaryotes are relatively recent and restricted to a single MC. The present analysis largely supports this interpretation, as the majority of candidates that have robust phylogenetic support for iHGT remain confined to a single MC. However, the expanded dataset also reveals a subset of candidates with broader taxonomic distributions or more complex phylogenetic patterns. In many of these cases, the inferred gene trees do not provide support for ancient horizontal transfer, as similar topologies can arise through differential gene loss, incomplete lineage sorting, or unresolved deep branching relationships. Consequently, distinguishing between ancient iHGT and vertical inheritance followed by gene loss remains challenging, particularly for genes that may trace back to early stages of eukaryotic evolution.

Importantly, the emergence of additional MCs for previously clade-restricted candidates raises a fundamental interpretative question: do these newly detected homologs reflect independent horizontal transfer events, ancient interdomain transfers near the origin of eukaryotes, or vertical inheritance followed by extensive differential gene loss? While all these scenarios are theoretically possible, vertical inheritance with lineage-specific loss provides the most parsimonious explanation in many cases. Interpreting such patterns as multiple independent iHGT events would require repeated transfers into distinct eukaryotic lineages, whereas invoking an ancient iHGT at the base of eukaryotes would additionally necessitate widespread and coordinated gene loss. By contrast, differential loss alone can generate similar presence–absence patterns without requiring improbable transfer dynamics.

This interpretation is consistent with recent analyses showing that the recovery of additional homologs can fundamentally shape phylogenetic signal, shifting the apparent origin of a gene toward the eukaryotic root and weakening support for iHGT (Aguirre-Carvajal and Armijos-Jaramillo, 2025). Moreover, theoretical work has demonstrated that gene loss alone can readily generate 'last-one-out' distributions, in which a gene appears restricted to a single lineage despite having been ancestrally present (Bremer et al., 2025). Such patterns can closely mimic recent horizontal acquisition, even in the absence of any transfer events.

These findings emphasize the limitations of relying exclusively on single-gene phylogenies to infer HGT. While gene trees remain a critical component of iHGT detection, anomalous topologies alone should not be interpreted as definitive evidence for transfer, particularly in deep evolutionary contexts. Integrative approaches that incorporate genomic context, taxon-specific homology searches, contamination screening, and comparative analyses across closely related species provide a more robust framework for evaluating candidate events. The revised methodology adopted here reflects this perspective and aims to reduce false positives arising from database incompleteness or technical artifacts.

Beyond the effects of taxon sampling, our results also reveal substantial method-dependent variability in iHGT inference. Automated and manual phylogenetic approaches differ in both the number of inferred iHGT events and the specific candidates they support, with only partial overlap between methods. Such discrepancies indicate that iHGT inference is highly sensitive to the criteria used to define sufficient phylogenetic evidence for horizontal transfer. Rather than reflecting methodological error, this variability highlights the intrinsic difficulty of interpreting phylogenetic signal in the presence of multiple, potentially confounding, evolutionary processes.

Interpreting putative iHGT events is particularly challenging because incongruent phylogenetic patterns can arise from several

alternative evolutionary scenarios, including incomplete lineage sorting, hybridization, introgression, gene loss, and recombination. These processes can either introduce or erase phylogenetic signal, thereby generating topologies that mimic horizontal transfer without involving any actual gene movement between distant lineages (Steenwyk et al., 2023). As a consequence, anomalous or patchy gene trees are not uniquely diagnostic of iHGT, especially when evaluated in isolation.

Consistent with this complexity, iHGT detection has long been recognized as lacking a definitive gold standard. Multiple studies have noted that different analytical approaches are expected to yield divergent results, an observation that remains valid today (Becq et al., 2010; Podell and Gaasterland, 2007). Early horizontal gene flow detection tools relied primarily on parametric signals such as deviations in GC content, codon usage or interpolated variable order motifs, which are now known to be insufficient on their own (Becq et al., 2010; Garcia-Vallve et al., 2003; Vernikos and Parkhill, 2006).

Subsequent methods incorporated implicit phylogenetic information derived from homology searches, including lineage probability indices and related scoring schemes (Boschetti et al., 2012; Gladyshev et al., 2008; Li et al., 2022; Podell and Gaasterland, 2007). In parallel, explicit phylogenetic reconciliation approaches sought to identify transfer events through discordance between gene trees and species trees (Bansal et al., 2018; David and Alm, 2011), although these methods have also been shown to generate false positives, even for conserved housekeeping genes (Dupont and Cox, 2017).

More recent frameworks, such as AVP (Koutsovoulos et al., 2022), attempt to integrate implicit and explicit strategies by combining homology-based metrics with phylogenetic evaluation while accounting for contamination-aware signals. While such integrative approaches represent an important methodological advance, our results demonstrate that even these frameworks remain sensitive to analytical assumptions and data context. Differences between AVP, Katz's original analysis and our manual evaluation therefore reflect not a single methodological failure, but rather the progressive incorporation of additional data, revised criteria and alternative evolutionary explanations.

Taken together, this methodological trajectory underscores that disagreement among iHGT detection approaches is an expected outcome of increasing analytical rigor, rather than a contradiction. As taxon sampling expands and interpretative frameworks become more conservative, the number of candidates with robust support for iHGT predictably decreases. This trend, observed in Katz (2015) and reinforced by the present analysis, reflects a growing awareness that many apparent iHGT signals dissolve under more comprehensive evolutionary scrutiny.

This study reinforces the view that iHGT into eukaryotes is neither ubiquitous nor negligible but instead represents a complex evolutionary process whose detection is highly sensitive to taxon sampling and methodological choices. As genomic databases continue to expand, previously proposed iHGT events –including those supported in the present analysis – will require ongoing reassessment. Rather than yielding a definitive count of interdomain transfers, such iterative analyses refine the boundaries within which robust inferences about HGT can be made.

## MATERIALS AND METHODS
### Phylogenetic reconstruction and iHGT inference
The present study uses as a starting dataset the 1138 putative iHGT candidates reported by Katz (2015), which were identified across multiple major eukaryotic clades, including Opisthokonta, SAR, Amoebozoa, Archaeplastida, Excavata, and Unclassified Eukaryota. In this work, all these candidates were evaluated to assess whether the iHGT hypotheses remain supported a decade later, using different detection methods, independent criteria and new genomic information currently available.

To perform taxonomy-specific homology searches, a DIAMOND v2.1.9 (Buchfink et al., 2021) database was constructed using the NCBI non-redundant (NR) protein dataset, together with NCBI taxonomic information obtained from the nodes.dmp and names.dmp files. A protein accession-to-taxonomy mapping file (prot.accession2taxid) was also incorporated. Each of these databases was accessed on December 3, 2025 release. The database was built using the following command: diamond makedb –in nr –taxonmap prot.accession2taxid –taxonnodes nodes.dmp –taxonnames names.dmp –db nr. This configuration enabled taxon-restricted homology searches using the –taxonlist option.

Candidate sequences were identified based on the alignments provided in the supplementary materials of Katz (2015), available through the Dryad repository (https://datadryad.org/dataset/doi:10.5061/dryad.2bj36). To ensure consistency with the original dataset, protein accession identifiers present in the alignments were extracted and used to retrieve the corresponding full-length sequences from NCBI, which were then used as queries in homology searches performed with DIAMOND. A complete list of query sequences used in the analysis is provided in Table S5.

To ensure a balanced and comparable homology search between prokaryotic and eukaryotic lineages, two separate taxonomy-restricted searches were conducted for each candidate using DIAMOND with the parameter –max-target-seqs 200. Prokaryote-specific searches were performed using –taxonlist 22157, corresponding to Bacteria (taxid: 2) and Archaea (taxid: 2157). Eukaryote-specific searches were performed using –taxonlist 2759, corresponding to Eukaryota.

Results from both taxon-specific searches were combined and filtered to retain only high-confidence homologs. Hits were excluded if they exhibited a pairwise identity below 30%, query coverage below 60%, or an E-value greater than $1\times10^{-5}$. Only the filtered homologs were used for downstream phylogenetic analyses.

Filtered homologous sequences were aligned using MAFFT v7.520 (Katoh and Standley, 2013) in automatic mode. Poorly aligned regions were subsequently removed using Gblocks v0.91b (Talavera and Castresana, 2007). The resulting alignments were converted to PHYLIP format, and ModelTest-NG v0.1.7 (Darriba et al., 2020) was used to determine the best-fitting amino acid substitution model for each dataset.

Phylogenetic trees were reconstructed using PhyML v3.3.20220408 (Guindon et al., 2010), with branch assessed using Shimodaira–Hasegawa (SH) support values. Final trees were annotated in NEXUS format using a custom Python v3.10.12 script that incorporated taxonomic information retrieved from GenBank records corresponding to the sequences in each phylogeny.

All phylogenetic trees were manually inspected using Geneious Prime 2025.2.1 (https://www.geneious.com; accessed 13 December 2025) to identify topological patterns consistent with potential iHGT events, following the patterns listed in Table 2.

### Evaluation of contamination
Additionally, a stringent contamination assessment was implemented for each candidate. This analysis evaluated both the genomic context of the candidate gene and the length of the scaffold on which it was located. For each candidate, the scaffold containing the query sequence was retrieved from the NCBI Nucleotide database (accessed 21 December 2025). The translated upstream and downstream flanking genes were then subjected to online NCBI BLASTp searches with default parameters (v2.13.0; accessed 21 December 2025) against the ClusteredNR database. These searches were used to assess whether the surrounding genomic context was consistent with the inferred donor lineage. Candidates whose flanking genes also matched the Prokaryota lineage were classified as Potential contamination. Conversely, candidates whose flanking regions did not support a donor-consistent genomic context were retained as likely genuine iHGT events.

### AVP iHGT detection
To compare manual phylogenetic inspection, AVP v1.0.10 (Koutsovoulos et al., 2022) was applied to all 1138 candidate sequences. AVP enables the

Biology Open

automatic identification of putative iHGT events using phylogeny-based approaches. The analysis begins with a preparation phase in which several metrics are computed for each candidate, including the Alien Index (AI) (Gladyshev et al., 2008), HGT index (Boschetti et al., 2012), Aggregate Hit Support (AHS) (Koutsovoulos et al., 2022), and the outgroup percentage (Li et al., 2022). Based on these metrics, an initial filtering step is applied: candidates with negative AI or AHS values and an outgroup percentage below 80 are classified as NO (no evidence of HGT).

Candidates that pass this filter proceed to the detection phase, where AVP performs phylogenetic tree inference and evaluates topological patterns to classify each candidate as iHGT, NO, Complex, or Unknown. For tree reconstruction, AVP supports either IQ-TREE or FastTree; in this study, IQ-TREE was used. AVP analyses require the candidate query sequence, the combined homology search results, and two taxonomic parameters: the ingroup and the exclusion group parameter (EGP). Detailed instructions for installing and running AVP are available in the AVP GitHub repository (https://github.com/GDKO/AvP).

The ingroup represents the taxonomic reference used to detect potential donors external to the group. Given that Katz focused on transfers from prokaryotes to eukaryotic MCs, Eukaryota was defined as the ingroup for all analyses. The EGP specifies the taxonomic group containing the candidate sequence and defines the lineage in which horizontal transfer is hypothesized to have occurred. The specific ingroup and EGP combinations used in this study are provided in Table S6. Outputs from AVP were compared with the manual phylogenetic detection pipeline (described above) to support a data-driven and conservative classification of candidates, allowing a robust determination of whether each case represents a supported iHGT event.

### Statistical assessment of taxon sampling effects on MC expansion

To evaluate whether the presence of specific taxonomic groups increases the likelihood that candidates proposed by Katz exhibit a higher number of MCs in our updated analysis, we relied on the presence–absence matrix of taxonomic groups reported in Katz's supplementary materials. This matrix was reannotated to match the taxonomic group definitions used in the present study (Table 1), reflecting updates and refinements to some categories relative to Katz (2015). We then conducted a statistical analysis based on logistic regression models to assess whether the inclusion of particular eukaryotic taxa is systematically associated with an increase in MC assignment.

A presence–absence matrix of the taxonomic groups identified in the analysis was generated using a custom Python script that employed the ETE v4 package (Huerta-Cepas et al., 2016) to retrieve and assign complete taxonomic information to each protein identifier. All statistical analyses were subsequently performed in RStudio v2022.7.2.576 using the stats package. Statistical significance was assessed using a two-sided threshold of 0.05.

The number of MCs in which each iHGT candidate was detected was quantified for both the original study and the present analysis. Based on this comparison, a binary response variable was defined to indicate whether the number of MCs increased in the updated analysis relative to the original classification.

Logistic regression models were fitted using the glm function with the argument family=binomial. The presence or absence of each analyzed eukaryotic taxon was considered a predictor variable, and models were fitted independently for each taxon.

For each logistic regression model, the P-value associated with the predictor coefficient was extracted and adjusted for multiple testing using the Bonferroni correction. In addition, predicted probabilities were estimated to quantify the effect of taxon presence on the likelihood of observing an increase in the number of identified MCs. These probabilities were computed by applying the inverse-logit transformation to the model coefficients, using the coef and plogis functions in R to obtain predicted values for the absence (intercept only) and presence (intercept plus predictor) of each taxonomic group.

To quantify differences in taxon sampling between the original study by Katz (2015) and our analysis, we estimated the number of species available for analysis in each taxonomic group under both study designs. Because our analysis was restricted to genomes with annotated proteomes, we focused on

estimating the number of species in NCBI that met these criteria rather than attempting to reconstruct exact historical database compositions.

For the updated dataset, species counts were estimated using the NCBI datasets command-line tool with the following parameters: datasets download genome [taxonomic group] –include protein –exclude-atypical –exclude-multi-isolate –preview.

The –preview option was used to retrieve genome metadata without downloading full genome assemblies, allowing efficient extraction of species-level information. Genomes flagged as atypical or derived from multiple isolates were excluded to reduce redundancy and potential biases associated with non-representative assemblies.

Species counts reported by Katz (2015) were obtained directly from the original publication and associated supplementary materials. For each taxonomic group, we then calculated the $\log_2$ fold change ($\log_2$ FC) in the number of species between the updated dataset and the Katz dataset. A pseudocount of 1 was added to avoid undefined values in cases where a taxonomic group was absent from either dataset.

In parallel, we computed the $\log_2$ fold change in the number of candidate genes detected per taxonomic group between the two studies using the same transformation. To assess whether changes in taxon sampling were associated with changes in candidate detection, we evaluated the correlation between $\log_2$ FC in species counts and $\log_2$ FC in candidate counts using Spearman's rank correlation coefficient, as implemented in R [cor.test(..., method="spearman")].

### Conclusions

This study visits the 1138 iHGT candidates originally reported by Katz (2015) by examining them under updated homology searches, expanded taxon sampling, and a combined automated and manual phylogenetic assessment framework. Rather than replicating the original methodology, we applied an independent criterion to evaluate the strength of phylogenetic support for prokaryote-to-eukaryote gene transfer under current database coverage. This approach allowed us to explore how differences in taxon representation and analytical assumptions influence iHGT inference.

Within the framework adopted here, about 30% of the evaluated candidates are consistent with interdomain horizontal transfer. The remaining candidates display phylogenetic patterns that lack support for prokaryote-to-eukaryote HGT and are compatible with alternative scenarios, including putative EGT, limited donor representation, absence of supporting homologs, or potential contamination. In many cases, the recovery of homologs from additional eukaryotic MCs results in broader phylogenetic distributions than previously documented, which reduces the strength of support for lineage-restricted transference.

Our analyses indicate that taxon sampling has a substantial influence on inferred MC distributions. Expanded genomic representation – particularly in Fungi, Metazoa, and Viridiplantae – is statistically associated with an increased likelihood of detecting additional eukaryotic homologs. Under the criteria applied in this study, the presence of such homologs often reduces support for interpretations based solely on apparent clade restriction. These results suggest that lineage-restricted distributions should be interpreted cautiously, particularly in the context of incomplete taxonomic sampling.

Taken together, our findings are consistent with the view that robust evidence for ancient iHGT in eukaryotes is limited under our phylogenetic inference. While some candidates remain compatible with relatively recent, clade-restricted transfer events, others exhibit broader or more ambiguous phylogenetic patterns when evaluated with expanded taxon sampling and different analytical thresholds. Overall, these results highlight the sensitivity of iHGT inference to database completeness and methodological assumptions and underscore the importance of iterative evaluation as genomic resources continue to expand.

### Acknowledgements
We are grateful to Helen Pugh for proofreading the manuscript. We thank the computational infrastructure of the Universidad de Las Américas for providing the resources that supported our analyses.

### Competing interests
The authors declare no competing or financial interests.

## Author contributions
Conceptualization: V.A.-J.; Data curation: K.A.-C., V.A.-J.; Formal analysis: K.A.-C.; Investigation: K.A.-C., V.A.-J.; Methodology: K.A.-C., V.A.-J.; Project administration: V.A.-J.; Resources: V.A.-J.; Writing – original draft: K.A.-C., V.A.-J.; Writing – review & editing: K.A.-C., V.A.-J.

## Funding
This research was funded by Universidad de Las Américas Ecuador as part of the program PRG.BIO.23.14.01. Open Access funding provided by Universidad de Las Américas Ecuador. Deposited in PMC for immediate release.

## Data and resource availability
All relevant data and details of resources can be found within the article and its supplementary information. Additional data are available in the Zenodo repository https://doi.org/10.5281/zenodo.18435322.

## First Person
This article has an associated First Person interview with the first author of the paper.

## Peer review history
The peer review history is available online at https://journals.biologists.com/bio/lookup/doi/10.1242/bio.062387.reviewer-comments.pdf

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
