## [Peer Review File · Biology Open]

What Impact Do New Homologs Have on Detecting Interdomain Horizontal Gene Transfer in Eukaryotes? A Reassessment of Katz (2015)

Kevin Aguirre-Carvajal and Vinicio Armijos-Jaramillo
DOI: 10.1242/bio.062387

Editor: Marie Monniaux

Review timeline

Original submission:	21 November 2025
Editorial decision:	2 December 2025
First revision received:	3 February 2026
Editorial decision:	16 February 2026
Second revision received:	23 February 2026
Accepted:	25 February 2026

Original submission

First decision letter

MS ID#: bio.062387

MS Title: What Impact Do New Homologs Have on Detecting Interdomain Horizontal Gene Transfer in Eukaryotes? A Reassessment of Katz (2015)

Authors: Kevin Aguirre-Carvajal; Vinicio Armijos-Jaramillo

I have now reached a decision on the above manuscript.

The reviewer reports are shown at the bottom of this email.

As you will see, the reviewers raised a number of substantial criticisms that prevent me from accepting the paper at this stage.

They suggest, however, that a revised version might prove acceptable, if you can address their concerns. If you think that you can deal satisfactorily with the criticisms on revision, I would be pleased to see a revised manuscript. We would then return it to the reviewers.

In particular, you will see that both reviewers have raised concerns about a possible flawed analysis:

(1) the homology search to detect prokaryotic homologs appears to be of an extremely limited depth, thereby preventing the full discovery of potential HGT events. Please provide additional elements to justify the methodology chosen;

(2) a large number of HGT events were discarded for various reasons that are not necessarily clear nor pertinent, which ends up supporting the initial assumption that HGT events are rare, in a circular reasoning. Please justify again, based on reviewer's comments, why these events were discarded.

(3) ensure that methods are described in sufficient details for greater reproducibility.

At this stage, we also ask you to ensure your manuscript complies with our formatting guidelines. Provided you are able to fully address the referees' comments, we are positive about publication of your paper (we accept over 95% of revision submissions) and therefore hope you won't mind any extra work involved in reformatting your manuscript at this point.

Please upload both a 'clean' version of your Word file, along with a highlighted version clearly showing where you have made changes in the revised manuscript. Please avoid using 'Track changes' in Word files as these are lost in PDF conversion.

I should be grateful if you would also provide a point-by-point response detailing how you have dealt with the points raised by the reviewers in the 'Response to Reviewers' box. Please attend to all of the reviewers' comments. If you do not agree with any of their criticisms or suggestions please explain clearly why this is so.

Reviewer 1

Comments for the author

Summary: As outlined below, it is my opinion that this manuscript by Aguirre-Carvajal & Armijos-Jaramillo: (1) has a critical flaw in its homolog search methodology, which renders the entire analysis difficult to interpret; (2) has insufficient methodological information in the text, and discrepancies between the text and the data, which make it impossible to reproduce the homolog search and the contaminant classification.

POINT 1: In response to "Are the manuscript's conclusions supported by the data?" I answered no.

The authors' central claim is that only 82 of the 638 HGT candidates from Katz (2015) stand up to modern scrutiny. Among the HGT candidates that the author claim to disprove, 119 were labelled "contaminants" and 192 were excluded because of "a lack of sufficient prokaryotic homologs in BLASTp searches." However, there appears to be a critical flaw in the methodology leading to the exclusion of some-or-all of these 311 HGT candidates: in lines 269-270 the authors describe that they used the "--max-target-seq 200" option in their DIAMOND homolog searches. So, the authors only sampled 200 hits from NR for each query.

This is an extremely small max target sequence cutoff. For reference, the publicly available NCBI server allows up to 10,000 targets w/ its BLASTP algorithm and 20,000 targets w/ its PSI-BLAST algorithm; and a search of NR can usually be completed on the NCBI server in seconds to minutes (and the authors used DIAMOND, which claims to be 100-10,000X faster than BLAST+).

This is particularly problematic for HGT candidates excluded because a prokaryotic homolog was not identified. It may simply be the case that the authors did not find prokaryotic homologs because of the very shallow depth of the homolog search. The authors even directly state this themselves in lines 119-123: "The absence of prokaryotic sequences in BLAST results is largely due to the overwhelming number of eukaryotic sequences displaying similarity to the query." Yet, these HGT candidates are seemingly excluded anyway, possibly falsely inflating the extent of the difference between this and Katz' analyses.

This setting is also potentially problematic for "contaminants," which the authors state were initially flagged "if its top BLAST hit belonged to a prokaryote or if the highest-scoring eukaryotic hit was followed exclusively by bacterial hits." By this, I presume the authors to mean the top hit from DIAMOND, because BLAST+ searches using "--max-target-seq n" returns the FIRST n results, not the TOP n results (see Shah et al 2018, Bioinformatics). Nevertheless, it was always my understanding that DIAMOND attempts to return the TOP n results when using "max-target-seq n," which would intuitively make this a good approach. However, upon review, I'm not certain this has actually been demonstrated (and I would welcome information from the authors on this). I quickly searched the DIAMOND github repository for more information, and found a comment from its

original creator and lead developer, stating: "if you use --max-target-seqs 1 in Diamond, it is designed to return the best hit, not just the first hit that meets the evaluate threshold ... there are heuristics involved however and there's no 100% guarantee that it will be the best hit" (<https://github.com/bbuchfink/diamond/issues/232>). Therefore, the probability that the authors may have erroneously recorded a prokaryotic hit as the "top hit" simply because of the very low max target sequence cutoff value seems too high to ignore. The authors did seem to use additional screening to define contaminants, but I did not get a full understanding of the criteria (see Point 2, below), so this is, at the very least, a point of great uncertainty.

The authors also excluded orphan HGT candidates (156-158), defining them as possible contaminants. If they were excluded simply because of their orphan status, and not because they were independently determined to be contaminants, this method of exclusion is in direct opposition to their claim that "for a fair comparison, we adhered to the criteria established in the original study to distinguish valid from invalid candidates" (Lines 72-74). Therefore, despite the magnitude of the difference being small, this falsely inflates the extent of the difference between this and Katz' analyses.

Given the extremely limited depth of the homology search, there is insufficient support for the authors central claim that "only 82 out of the 683 originally proposed HGT candidates (involving a single major clade) [remain] valid" (Lines 181-182). It is my opinion that *exhaustive* searches are required to sufficiently support the authors claims (and this does not seem unreasonable, given the speed at which this can be done on public databases, using free online tools). Unfortunately, this would also mean that all downstream analyses would have to be repeated.

POINT 2: In response to "Does the methods section provide sufficient detail to permit reproducibility?" I answered "no."

It is not clear that the max-target-seq cutoff value reported is accurate. I glanced at the multiple sequence alignments and found that many have more than the 1-200 (or 1-201, if including the query) sequences expected from a search using a max-target-seq cutoff of 200. A random set of 3 examples: OG5_127635_XP_002423766.1_align.fasta-gb [203 seqs]; OG5_146455_XP_040528810.1_align.fasta-gb [473 seqs]; and OG5_141276_SCO61246.1_align.fasta-gb [255 seqs]. I do not understand this discrepancy, which leads me to believe that there is insufficient information or some mistake in the methods.

Also, the authors do not give a complete description of their methods for defining putative contaminants. The authors state that, after flagging potential contaminants, "To refine the dataset further, additional analyses included retrieving the contig of the query sequence and conducting NCBI Online BLASTp (v2.13.0, accessed on March 2th [sic], 2025; <https://blast.ncbi.nlm.nih.gov/>), searches on upstream and downstream translated genes to assess whether surrounding genes were of bacterial origin." It is not clear: (1) the exact criteria for exclusion -- i.e., what did the authors need to find to exclude a sequence as contamination?; (2) the settings for these searches; (3) why they specific bacterial, and not more generally prokaryotic (perhaps a typo); and (4) if all 119 "contaminant" exclusions met the same criteria.

Finally, the authors state that they used "DIAMOND BLAST" for their initial homolog searches. This is presumably a minor text error, as DIAMOND and BLAST are standalone programs (although DIAMOND can interface with and output BLAST-compatible formats). However, throughout, the authors reference things like "BLAST search results." I presume they used DIAMOND throughout, but this isn't sufficiently clear to me, as the authors clearly also used the NCBI BLAST+ suite for contaminant classification. The exact programs and settings need to be made clear throughout. If the authors in fact did use BLAST+, it's worth reiterating that the problems outlined above with "max-target-seq" are potentially more extreme, as BLAST+ with "--max-target-seq n" returns the FIRST n results not the TOP n results (see Shah et al 2018, Bioinformatics).

Reviewer 2

Comments for the author

I have read this manuscript with pleasure and interest. The authors revisit key findings in a classic paper in the field of comparative genomics of lateral gene transfer events (Katz PRSB 2015) and critically assess their validity in light of updated datasets and methods, and discuss possible implications for our understanding of eukaryotic genome evolution.

A major criticism of this work is that many of their methodological choices seem to be motivated by the unstated assumption that LGT is rare, which is precisely what the authors are trying to assess. Thus, these choices are affected by circular reasoning. For example, the authors chose not to consider candidate LGT events from Katz 2015 if they involved genes from more than one major eukaryotic clade on the basis that assuming LGT events 'near the eukaryotic root' is not parsimonious, but offer no basis for this. It is true that LGT and vertical inheritance events during eukaryogenesis, followed by multiple independent losses, would create a very similar presence/absence pattern of that gene across extant eukaryotic genomes. However, the authors should be aware of the fact that during the eukaryogenesis process many genes from various prokaryotic donors (bacterial and archaeal) became integrated in a single symbiotic genome (FECA-to-LECA) at different stages and that the distinction between LGT and vertical inheritance in this context is moot or impossible to specify (cf 10.5802/crbiol.118 and papers therein). Distinguishing between these two scenarios necessitates investigating the detailed topologies of the gene phylogenies, which is only partially covered in this study.

Likewise, from L318: If I understand this correctly, LGT candidates present in more than one eukaryotic major group are automatically excluded, is that correct? This criterion is new to this study and contradicts those of Katz 2015, which did consider the possibility of HGT events involving up to three eukaryotic major groups. This is in contradiction with the authors' statement that they adhered to the same criteria as in Katz 2015 (L72), something that is stated in more than one instance in the manuscript and should be amended. This is an important drawback in the authors' methodology because this appears to be the single most abundant reason why a candidate LGT events gets discarded, according to Table 2 (272 cases). An explicit assessment of which and how many clades have been detected in addition to the single original one, and whether their addition is phylogenetically congruent or not, would be welcome. For example, the detection of Cryptophyta homologs of previously Archaeplastida-only candidate LGT genes is less incongruent with LGT than the detection of new homologs in a phylogenetically diverse range of clades.

Other comments:

L114 - which criteria did the authors use to classify an HGT event as "inconclusive"? Clarify.

L119, L270 – The authors rely on a fixed number of top hits to decide if there are sufficient (how many?) prokaryotic hits for any given query sequence. Given that databases have grown in size over the past decade, this is surely going to bias the results in unpredictable ways, especially if the authors consider only 200 top hits (L270). It would have been more robust to filter the hit list based on alignment scores (bitscores) or E-values rather than the sheer number of hits. This is a major source of concern because 192 out of 683 candidate LGTs could not be assessed because of this, according to Table 2.

L189 – The conclusions from these two papers should be explained succinctly.

L220 – Note that the statement in Katz 2015 was motivated by the fact that, back then, the only available excavates were highly reduced parasitic genomes, and only one rhizarian was sampled. These extreme situations in sampling bias are not accounted for in your analysis of database effect on LGT detection.

L247 – Here and elsewhere in the paper, the authors state that relying solely on gene trees for LGT analysis could be "premature" or otherwise misguided. This discussion section is a good place to put forward alternative methods.

General comment on the discussion – a key finding by Katz 2015 is that most of the interdomain LGT events there described were relatively recent (restricted to one major eukaryotic clade). It would be worth discussing how many of these are now inferred to be older.

L281 - Please provide more details on how AVP was used for this analysis. Related to this: did the authors use IQTREE (L282) or Phylml (L301) for their gene tree inferences? Please clarify.

Comments on figures:

Fig. 1 – The legibility of this figure would be much improved if the authors expanded their colour-coding to distinguish between prokaryotic and eukaryotic genes more generally.

Reviewer's Responses to Questions

Experimental quality

Does each figure have the proper controls?

If 'No', please indicate reasons in Comments for Author box below.

Reviewer #1:

- Yes

Reviewer #2:

- Yes

Were the data analyzed using appropriate statistical tests?

If 'No', please indicate reasons in Comments for Author box below.

Reviewer #1:

- Yes

Reviewer #2:

- Yes

Reproducibility

Were experiments performed using adequate number of biological replicates?

If 'No', please indicate reasons in Comments for Author box below.

Reviewer #1:

- Yes

Reviewer #2:

- Yes

Does the methods section provide sufficient detail to permit reproducibility?

If 'No', please indicate reasons in Comments for Author box below.

Reviewer #1:

- No

Reviewer #2:

- No

Completeness

Are the manuscript's conclusions supported by the data?

If 'No', please indicate reasons in Comments for Author box below.

Reviewer #1:

- No

Reviewer #2:

- No

Scholarship

Do the authors cite and discuss the merits of data that would argue for and against their conclusion?

If 'No', please indicate reasons in Comments for Author box below.

Reviewer #1:

- Yes

Reviewer #2:

- No

Does the manuscript title & abstract accurately reflect the contents of the manuscript, without hyperbole?

If 'No', please indicate reasons in Comments for Author box below.

Reviewer #1:

- Yes

Reviewer #2:

- Yes

First revision

Author response to reviewers' comments

Review 1

We thank the reviewer for these valuable observations, which led to substantial improvements and strengthened the overall rigor of the manuscript.

(1) has a critical flaw in its homolog search methodology, which renders the entire analysis difficult to interpret.

Concern 1.1: “The authors' central claim is that only 82 of the 638 HGT candidates from Katz (2015) stand up to modern scrutiny. Among the HGT candidates that the author claim to disprove, 119 were labelled “contaminants” and 192 were excluded because of “a lack of sufficient prokaryotic homologs in BLASTp searches.” However, there appears to be a critical flaw in the methodology leading to the exclusion of some-or-all of these 311 HGT candidates: in lines 269-270 the authors describe that they used the “--max-target-seq 200” option in their DIAMOND homolog searches. So, the authors only sampled 200 hits from NR for each query. This is an extremely small max target sequence cutoff. For reference, the publicly available NCBI server allows up to 10,000 targets w/ its BLASTP algorithm and 20,000 targets w/ its PSI-BLAST algorithm; and a search of NR can usually be completed on the NCBI server in seconds to minutes (and the authors used DIAMOND, which claims to be 100-10,000X faster than BLAST+).”

We thank the reviewer for raising this concern. We acknowledge the potential bias associated with considering only the first 200 database hits. In the revised version of the manuscript, we therefore include the top 200 eukaryotic hits and the top 200 prokaryotic hits for each candidate, ensuring that the most similar homologs from each group are represented. The reviewer's concern regarding the use of the --max-target-seqs 200 parameter in DIAMOND arises from a misunderstanding of how this option operates. This parameter does not limit the search to only 200 sequences from the database, nor does it restrict the search space. Instead, DIAMOND compares the query against all

sequences in the database, ranks all matches by similarity score, and subsequently reports only the top 200 highest-scoring hits.

Importantly, this behavior differs conceptually from sampling or truncating the database prior to the search. The homology search therefore remains exhaustive with respect to the database content, and the parameter only affects the number of reported results retained for downstream analyses.

Furthermore, the chosen cutoff represents a more permissive setting than DIAMOND's default, which reports only 25 target sequences by default as is described in the documentation (<https://github.com/bbuchfink/diamond/wiki/3.-Command-line-options>). Thus, increasing `--max-target-seqs` to 200 substantially expands the number of retained homologs relative to default behavior, ensuring that distant but relevant prokaryotic homologs are captured when present. This choice balances sensitivity with computational efficiency while avoiding the excessive inclusion of weak, low-confidence hits that may complicate phylogenetic interpretation.

The methodological decision to classify certain candidates as lacking sufficient prokaryotic homologs was therefore not driven by an artificial limitation of the search space, but rather reflects the genuine absence of robust bacterial homologs among the highest-confidence matches across the database.

Concern 1.2: *“This is particularly problematic for HGT candidates excluded because a prokaryotic homolog was not identified. It may simply be the case that the authors did not find prokaryotic homologs because of the very shallow depth of the homolog search. The authors even directly state this themselves in lines 119-123: “The absence of prokaryotic sequences in BLAST results is largely due to the overwhelming number of eukaryotic sequences displaying similarity to the query.” Yet, these HGT candidates are seemingly excluded anyway, possibly falsely inflating the extent of the difference between this and Katz’ analyses.”*

The reviewer raises a valid point regarding the potential bias introduced by a single, unrestricted homology search when eukaryotic sequences dominate the top-ranking hits. As noted in the manuscript, the absence of detectable prokaryotic homologs in some cases may indeed result from the overwhelming representation of eukaryotic sequences with high similarity to the query, rather than from a true lack of prokaryotic homologs.

To address this concern and avoid the possible under-detection of prokaryotic homologs, the complete phylogenetic analysis was repeated using a revised homology search strategy. Specifically, two independent DIAMOND searches were conducted for each candidate: one restricted to eukaryotic sequences and a second restricted to prokaryotic sequences using the `--taxonlist` parameter (Line 416-426). This approach ensures balanced and explicit sampling of homologs from both domains, independent of their relative abundance or similarity rankings in an unrestricted search.

The resulting homolog sets from both searches were then combined prior to downstream filtering, alignment, and phylogenetic reconstruction. We opted to reevaluate the complete candidates, the 1138, for this revision. Candidates were reassessed based on these expanded and taxonomically controlled datasets, thereby minimizing the risk of falsely excluding HGT candidates due to biased homolog recovery. This revised strategy provides a more conservative and methodologically robust framework for evaluating the presence or absence of prokaryotic homologs and strengthens the reliability of the reported differences relative to Katz (2015).

Concern 1.3: *“This setting is also potentially problematic for “contaminants,” which the authors state were initially flagged “if its top BLAST hit belonged to a prokaryote or if the highest-scoring eukaryotic hit was followed exclusively by bacterial hits.” By this, I presume the authors to mean the top hit from DIAMOND, because BLAST+ searches using “--max-target-seq n” returns the FIRST n results, not the TOP n results (see Shah et al 2018, Bioinformatics). Nevertheless, it was always my understanding that DIAMOND attempts to return the TOP n*

results when using "max-target-seq n," which would intuitively make this a good approach. However, upon review, I'm not certain this has actually been demonstrated (and I would welcome information from the authors on this). I quickly searched the DIAMOND github repository for more information, and found a comment from its original creator and lead developer, stating: "if you use --max-target-seqs 1 in Diamond, it is designed to return the best hit, not just the first hit that meets the evaluate threshold ... there are heuristics involved however and there's no 100% guarantee that it will be the best hit" (<https://github.com/bbuchfink/diamond/issues/232>). Therefore, the probability that the authors may have erroneously recorded a prokaryotic hit as the "top hit" simply because of the very low max target sequence cutoff value seems too high to ignore. The authors did seem to use additional screening to define contaminants, but I did not get a full understanding of the criteria (see Point 2, below), so this is, at the very least, a point of great uncertainty. The authors also excluded orphan HGT candidates (156-158), defining them as possible contaminants. If they were excluded simply because of their orphan status, and not because they were independently determined to be contaminants, this method of exclusion is in direct opposition to their claim that "for a fair comparison, we adhered to the criteria established in the original study to distinguish valid from invalid candidates" (Lines 72-74). Therefore, despite the magnitude of the difference being small, this falsely inflates the extent of the difference between this and Katz' analyses."

The concern correctly highlights two related issues: (i) the interpretation of "top hits" returned by DIAMOND when using the --max-target-seqs parameter, and (ii) the criteria used to classify candidates as potential contaminants, including the treatment of orphan candidates.

Regarding the first point, we confirm that DIAMOND is explicitly designed to prioritize the reporting of best-scoring hits rather than the first hits satisfying an E-value threshold when using --max-target-seqs. As noted by the developer, even when --max-target-seqs is set to low values, DIAMOND aims to return the best hit rather than an arbitrary early match (<https://github.com/bbuchfink/diamond/issues/232>).

Importantly, this behavior is supported by DIAMOND's published algorithmic framework. DIAMOND employs an *adaptive ranking* strategy in which target sequences are initially ordered based on ungapped extension scores derived from seed hits, providing a preliminary approximation of alignment quality. Target sequences are then processed in ranked chunks, and gapped extensions are computed dynamically until no further significant alignments are detected under the specified reporting criteria (Buchfink et al., 2021). This adaptive procedure improves reporting accuracy relative to static cutoffs by focusing computational effort on the most promising candidates while reducing ranking bias.

Nevertheless, we acknowledge—as also stated by the DIAMOND developers—that heuristic approximations are involved and that there is no absolute guarantee that the globally optimal hit is always returned, particularly when --max-target-seqs is set to low values. We therefore agree with the reviewer that contamination assessment based solely on top-hit identity can introduce uncertainty.

With respect to contamination detection, we fully agree that the original criteria allowed room for improvement and may have contributed to ambiguity. In response, we substantially revised and strengthened the contamination assessment pipeline. In the revised analysis, candidates are no longer classified as potential contaminants based on orphan status or solely on the taxonomic identity of top-ranked homologs. We acknowledge that treating orphan HGT candidates as potential contaminants could artificially inflate this category and does not align with the stated goal of adhering to the criteria of the original study.

Instead, contamination assessment is now based exclusively on genomic context. For each candidate, the scaffold or contig containing the query sequence was retrieved, and the taxonomic affiliations of flanking genes were examined. Candidates were classified as potential contaminants only when flanking genes consistently matched the same putative donor lineage, supporting a donor-consistent genomic context indicative of contamination. Conversely, candidates whose flanking regions did not support such a pattern were retained as likely genuine HGT events. This

more stringent and conservative criterion reduces false-positive contamination assignments and avoids penalizing orphan candidates in the absence of independent evidence for contamination.

The revised contamination assessment procedure is now described more clearly and concisely in the Methods section (Lines 461-474 of the revised manuscript). These changes ensure that contamination classification is independent of homolog search depth, orphan status, or top-hit uncertainty, thereby minimizing the risk of artificially inflating the differences between our results and the original research (Katz, 2015).

Concern 1.4: *“Given the extremely limited depth of the homology search, there is insufficient support for the authors central claim that “only 82 out of the 683 originally proposed HGT candidates (involving a single major clade) [remain] valid” (Lines 181-182). It is my opinion that *exhaustive* searches are required to sufficiently support the authors claims (and this does not seem unreasonable, given the speed at which this can be done on public databases, using free online tools). Unfortunately, this would also mean that all downstream analyses would have to be repeated.”*

We agree that robust inference regarding the validity of HGT candidates requires comprehensive sampling of homologs, and we acknowledge that insufficient search depth could undermine downstream analyses.

As noted by the reviewer, this concern is closely related to Concern 1.2, and we have addressed it by substantially revising our homology search strategy rather than relying on a single, shallow search. Specifically, we re-ran the complete phylogenetic pipeline using two complementary, taxon-specific homology searches: one restricted to Eukaryota and one restricted to Prokaryota, implemented using DIAMOND with the --taxon-list option. This approach ensures exhaustive recovery of homologs from both domains independently, thereby avoiding competitive bias caused by the overwhelming abundance of prokaryotic sequences in public databases.

Importantly, this revision required repeating the entire downstream analytical pipeline. As the homology search strategy was fundamentally redesigned, all 1,138 candidates originally reported by Katz were re-evaluated de novo, including homolog retrieval, multiple sequence alignment, phylogenetic reconstruction, manual tree inspection and the AVP tool pipeline. Consequently, the central claim of the manuscript—that only a subset of the originally proposed Katz candidates remains supported under updated and taxon-balanced sampling—relies exclusively on these revised analyses rather than on the initial single-search strategy.

Additionally, this reanalysis highlights the central role of database completeness in HGT detection. Phylogeny-based approaches rely strongly on the availability of homologous sequences, and more comprehensive databases increase the likelihood of recovering homologs relevant to candidate genes.

(2) has insufficient methodological information in the text, and discrepancies between the text and the data, which make it impossible to reproduce the homolog search and the contaminant classification.

Concern2.1: *“It is not clear that the max-target-seq cutoff value reported is accurate. I glanced at the multiple sequence alignments and found that many have more than the 1-200 (or 1-201, if including the query) sequences expected from a search using a max-target-seq cutoff of 200. A random set of 3 examples: OG5_127635_XP_002423766.1_align.fasta-gb [203 seqs]; OG5_146455_XP_040528810.1_align.fasta-gb [473 seqs]; and OG5_141276_SCO61246.1_align.fasta-gb [255 seqs]. I do not understand this discrepancy, which leads me to believe that there is insufficient information or some mistake in the methods.”*

Following the reviewer’s concern, we reformulate the entire methodology section and we repeated the entire phylogenetic analysis from scratch. This reanalysis included all 1,138 Katz candidates and encompassed homolog retrieval, multiple sequence alignment, phylogenetic reconstruction, and manual tree evaluation. During this process, particular care was taken to ensure consistency between the methods described in the text and the data provided.

The revised Methods section now provides a substantially more detailed and explicit description of the homolog retrieval strategy. Specifically, we clarify how DIAMOND databases were constructed for taxon-specific searches (Lines 416-426), how separate eukaryotic and prokaryotic homology searches were conducted, and which parameters were used for the --taxon-list option (Lines 438-441). We also explicitly describe how homologs from different searches were combined prior to alignment construction (Line 442-443). Our phylogenetic pipeline was reformulated and it is explained in the lines 442-460. The contamination assessment description was improved and described in the lines 461-474. AVP tool is detailed in lines 475-503 and also downstream statistical analysis in lines 504-562.

These revisions eliminate the discrepancy noted by the reviewer and ensure that the homolog search strategy and downstream analyses are fully transparent and reproducible.

Concern2.2: *“Also, the authors do not give a complete description of their methods for defining putative contaminants. The authors state that, after flagging potential contaminants, “To refine the dataset further, additional analyses included retrieving the contig of the query sequence and conducting NCBI Online BLASTp (v2.13.0, accessed on March 2th [sic], 2025; <https://blast.ncbi.nlm.nih.gov/>), searches on upstream and downstream translated genes to assess whether surrounding genes were of bacterial origin.” It is not clear: (1) the exact criteria for exclusion -- i.e., what did the authors need to find to exclude a sequence as contamination?; (2) the settings for these searches; (3) why they specific bacterial, and not more generally prokaryotic (perhaps a typo); and (4) if all 119 “contaminant” exclusions met the same criteria.”*

We thank the reviewer for pointing out that the original manuscript did not provide sufficient detail regarding the contamination assessment procedure. We agree that the previous description lacked clarity and reproducibility. In response, we substantially revised and clarified this section, reformulated the contamination criteria, and applied a stricter and more conservative framework consistently across all candidates.

(1) Exact criteria for exclusion

In the revised analysis, candidates were classified as potential contaminants *only* when independent genomic-context evidence supported contamination. Specifically, a candidate was excluded as contamination when the upstream and downstream flanking genes located on the same scaffold consistently showed strong homology to prokaryotic lineages, indicating a donor-consistent genomic context rather than an isolated horizontal transfer event (Lines 471-474).

Candidates were *not* excluded based on orphan status, top-hit identity, or homology patterns alone. If the flanking regions did not support a prokaryotic genomic context, the candidate was retained as a likely genuine HGT event.

(2) Search settings

To ensure reproducibility, the revised Methods section now explicitly states that translated flanking genes were queried using online NCBI BLASTp (v2.13.0) with default parameters against the ClusteredNR database. No additional filters or taxonomic constraints were applied during these searches. This information is now clearly documented in the Methods section (Lines 464-469).

(3) Use of “bacterial” vs. “prokaryotic” terminology

We agree with the reviewer that the use of the term “bacterial” in the original text was imprecise. This has been corrected in the revised manuscript to “prokaryotic,” reflecting the actual scope of the contamination assessment. The revised text consistently refers to prokaryotic lineages (including both Bacteria and Archaea) to avoid ambiguity. For example, in the methodology section in line 471 we refer to Prokaryota instead of bacteria.

(4) Consistency across all contaminant exclusions

All candidates classified as potential contaminants in the revised dataset were evaluated using the same contamination assessment criteria described above. No candidate was excluded using alternative or ad hoc rules. Importantly, following this revision, orphan status alone was no longer considered a valid basis for contamination classification, which reduced the risk of inflating the contamination category and ensured consistent application of criteria across all candidates.

This genomic-context-based contamination assessment follows established practice in HGT studies and is not unique to this work. Similar approaches relying on scaffold composition and flanking gene identity have been used to distinguish bona fide HGT events from sequencing contamination (Ma et al., 2022). By adopting this conservative strategy, contamination classification is now independent of homology search depth and ranking heuristics.

The revised contamination assessment procedure is now described in detail in the Methods section (Lines 461-474), ensuring transparency, reproducibility, and consistency across all candidates.

Concern2.3: Finally, the authors state that they used "DIAMOND BLAST" for their initial homolog searches. This is presumably a minor text error, as DIAMOND and BLAST are standalone programs (although DIAMOND can interface with and output BLAST-compatible formats). However, throughout, the authors reference things like "BLAST search results." I presume they used DIAMOND throughout, but this isn't sufficiently clear to me, as the authors clearly also used the NCBI BLAST+ suite for contaminant classification. The exact programs and settings need to be made clear throughout. If the authors in fact did use BLAST+, it's worth reiterating that the problems outlined above with "max-target-seq" are potentially more extreme, as BLAST+ with "--max-target-seq n" returns the FIRST n results not the TOP n results (see Shah et al 2018, Bioinformatics).

We agree that the original wording could lead to confusion. In the initial version of the manuscript, the phrase "DIAMOND BLAST" was used to indicate that DIAMOND was employed to perform sequence homology searches using a BLAST-like local alignment algorithm. We acknowledge that this phrasing was imprecise and potentially misleading. DIAMOND and NCBI BLAST+ are distinct software packages; however, DIAMOND is designed to produce BLAST-compatible output formats and implements a BLAST-like alignment strategy that is optimized for speed. In the revised manuscript, this terminology has been corrected to clearly and consistently distinguish between DIAMOND and BLAST+. Notably, the term "BLAST" now appears only once in the Methods section (Line 467), where it explicitly refers to the use of NCBI BLASTp during the contamination assessment step.

To clarify tool usage:

- **DIAMOND** was used exclusively for the initial large-scale homolog retrieval.
- **ONLINE NCBI BLAST+ (BLASTp)** was used only during the contamination assessment step, specifically to evaluate the taxonomic identity of translated upstream and downstream flanking genes.

No BLAST+ searches were used for primary homolog discovery or for defining the core phylogenetic datasets.

Importantly, the `--max-target-seqs` parameter was applied **only in DIAMOND searches**, not in BLAST+. As noted by the reviewer, BLAST+ with `--max-target-seqs n` may return the *first* n hits rather than the *best* n hits (Shah et al., 2019). However, this limitation does not apply to our BLAST+ usage, as BLAST+ was not used for ranked homolog retrieval or candidate exclusion based on hit order.

We further note that the concerns raised by Shah et al. (2019) regarding `--max-target-seqs` were subsequently addressed in the official response, which confirm the bug and fix it since version BLAST+ 2.8.1 (Madden et al., 2019). Nonetheless, to avoid any ambiguity, we have ensured that

ranked homology inference relied solely on DIAMOND, while BLAST+ was restricted to qualitative contamination checks based on genomic context.

The revised Methods section now clearly specifies the software, versions, parameters, and purpose of each tool at every analysis stage, eliminating ambiguity and ensuring reproducibility.

References

- Buchfink, B., Reuter, K., & Drost, H.-G. (2021). Sensitive protein alignments at tree-of-life scale using DIAMOND. *Nature Methods*, 18(4), 366-368. <https://doi.org/10.1038/s41592-021-01101-x>
- Katz, L. A. (2015). Recent events dominate interdomain lateral gene transfers between prokaryotes and eukaryotes and, with the exception of endosymbiotic gene transfers, few ancient transfer events persist. *Philosophical Transactions of the Royal Society B: Biological Sciences*, 370(1678), 20140324. <https://doi.org/10.1098/rstb.2014.0324>
- Ma, J., Wang, S., Zhu, X., Sun, G., Chang, G., Li, L., Hu, X., Zhang, S., Zhou, Y., Song, C.-P., & Huang, J. (2022). Major episodes of horizontal gene transfer drove the evolution of land plants. *Molecular Plant*, 15(5), 857-871. <https://doi.org/10.1016/j.molp.2022.02.001>
- Madden, T. L., Busby, B., & Ye, J. (2019). Reply to the paper: Misunderstood parameters of NCBI BLAST impacts the correctness of bioinformatics workflows. *Bioinformatics*, 35(15), 2699-2700. <https://doi.org/10.1093/bioinformatics/bty1026>
- Shah, N., Nute, M. G., Warnow, T., & Pop, M. (2019). Misunderstood parameter of NCBI BLAST impacts the correctness of bioinformatics workflows. *Bioinformatics (Oxford, England)*, 35(9), 1613-1614. <https://doi.org/10.1093/bioinformatics/bty833>

Review 2

We thank the reviewer for these valuable observations, which led to substantial improvements and strengthened the overall rigor of the manuscript.

Concern 1.1: “A major criticism of this work is that many of their methodological choices seem to be motivated by the unstated assumption that LGT is rare, which is precisely what the authors are trying to assess. Thus, these choices are affected by circular reasoning. For example, the authors chose not to consider candidate LGT events from Katz 2015 if they involved genes from more than one major eukaryotic clade on the basis that assuming LGT events ‘near the eukaryotic root’ is not parsimonious, but offer no basis for this. It is true that LGT and vertical inheritance events during eukaryogenesis, followed by multiple independent losses, would create a very similar presence/absence pattern of that gene across extant eukaryotic genomes. However, the authors should be aware of the fact that during the eukaryogenesis process many genes from various prokaryotic donors (bacterial and archaeal) became integrated in a single symbiotic genome (FECA-to-LECA) at different stages and that the distinction between LGT and vertical inheritance in this context is moot or impossible to specify (cf 10.5802/crbiol.118 and papers therein). Distinguishing between these two scenarios necessitates investigating the detailed topologies of the gene phylogenies, which is only partially covered in this study.”

We acknowledge that some of the assumptions in the original version of the manuscript could be perceived as introducing bias toward identifying fewer LGT events, potentially giving the impression of circular reasoning. However, the observed reduction in inferred HGT cases consistently emerges when previously reported candidates are reanalyzed, regardless of the specific methodology applied. For instance, in the present reanalysis, the AVP tool—an approach independent of the a priori criteria used in our manual pipeline—also recovers a substantially lower number of HGT candidates. This concordant outcome suggests that the tendency to detect fewer HGT events is not driven by a particular analytical framework, but rather reflects a broader pattern across methods. In the revised manuscript, we have implemented notable methodological changes, yet the same trend persists: reanalyzes systematically yield fewer HGT cases compared to the original reports. We therefore use this observation as a basis to discuss several alternative explanations that may account for this recurring pattern.

In the original version of the manuscript, the treatment of candidates present in multiple eukaryotic major clades (MCs) was partly motivated by a parsimony-based argument, which the reviewer correctly notes requires careful justification. We agree that gene presence across multiple MCs can arise from several evolutionary scenarios, including early LGT events followed by vertical inheritance and lineage-specific losses, and that presence/absence patterns alone are insufficient to discriminate among these possibilities.

In response to this concern, we revised the study design and reanalyzed all 1,138 candidates originally reported by Katz (Line 412-415), irrespective of the number of eukaryotic major clades involved. Importantly, candidate evaluation is no longer based on MC counts per se, but instead relies on detailed phylogenetic tree inspection (Line 457-460), contamination assessment based on genomic context (Line 461-474) and automated phylogeny-based inference using AVP (Line 475-503). This approach minimizes circular assumptions regarding the frequency or timing of LGT events.

We emphasize that candidates were not excluded simply because they involved multiple eukaryotic MCs. Rather, such candidates were classified as inconclusive when phylogenetic topologies were compatible with multiple evolutionary scenarios (e.g., early acquisition followed by differential loss, incomplete lineage sorting, or repeated transfers) and therefore lacked a robust, unambiguous HGT signal.

The following illustration (Fig 1) shows alternative evolutionary scenarios that can give rise to identical extant gene distributions. In Alternative 1, the gene is inferred to have been present in the ancestral eukaryotic lineage, followed by multiple independent losses in Amoebozoa and SAR. In Alternative 2, the same pattern is explained by a horizontal gene transfer (HGT) event from bacteria into the stem lineage of Excavata, followed by subsequent losses in other eukaryotic groups. Although both scenarios can, in principle, account for the observed distribution, they differ in the number and nature of evolutionary events required. When gene tree topology recovers a monophyletic eukaryotic clade that branches sister to bacteria, vertical inheritance followed by differential gene loss provides a parsimonious explanation that does not require invoking an additional interdomain transfer event. Importantly, this reasoning does not presuppose that HGT is rare, nor does it exclude early prokaryotic gene acquisition during eukaryogenesis. Rather, it reflects the fact that once a gene is inferred to be present in the ancestral eukaryotic lineage, the observed absence in specific clades can be explained by loss alone, whereas an HGT-based scenario necessarily requires both acquisition and loss events to reproduce the same pattern. Under such conditions, gene loss represents a sufficient and parsimonious explanation, and the available phylogenetic signal does not allow these cases to be confidently classified as horizontal transfer events. We also included this topic in discussion (Line 319-338).

Fig. 1. Alternative evolutionary scenarios leading to complex phylogenetic patterns that may be interpreted as HGT-like topologies. Refer to Figure 2 of Aguirre-Carvajal & Armijos-Jaramillo (2025) (<https://onlinelibrary.wiley.com/doi/pdf/10.1002/ece3.72653>).

We agree with the reviewer that distinguishing between early LGT and vertical inheritance during eukaryogenesis is often intrinsically difficult or impossible based solely on extant data. Accordingly, our conclusions are intentionally conservative and restricted to cases where phylogenetic evidence provides strong support for HGT under current taxon sampling, rather than making claims about the absolute frequency of LGT during early eukaryotic evolution.

Concer 2.1: “Likewise, from L318: If I understand this correctly, LGT candidates present in more than one eukaryotic major group are automatically excluded, is that correct? This criterion is new to this study and contradicts those of Katz 2015, which did consider the possibility of HGT events involving up to three eukaryotic major groups. This is in contradiction with the authors’ statement that they adhered to the same criteria as in Katz 2015 (L72), something that is stated in more than one instance in the manuscript and should be amended. This is an important drawback in the authors’ methodology because this appears to be the single most abundant reason why a candidate LGT events gets discarded, according to Table 2 (272 cases). An explicit assessment of which and how many clades have been detected in addition to the single original one, and whether their addition is phylogenetically congruent or not, would be welcome. For example, the detection of Cryptophyta homologs of previously Archaeplastida-only candidate LGT genes is less incongruent with LGT than the detection of new homologs in a phylogenetically diverse range of clades.”

In the original version, the manuscript incorrectly stated that we “applied the same criteria as in Katz (2015)”. We agree with the reviewer that this statement was inaccurate and potentially misleading. While Katz (2015) explicitly allowed candidate HGT events to involve up to three major eukaryotic clades, our initial filtering strategy differed in that candidates detected in more than one major eukaryotic clade were provisionally treated with greater caution, as they could reflect

alternative evolutionary scenarios (e.g., vertical inheritance with differential loss or early gene acquisition).

Importantly, and as clarified in response to Concern 1.1, we did not exclude such candidates a priori from the reanalysis in this version. Instead, we reanalyzed all 1,138 original candidates, irrespective of the number of major clades detected in the updated searches. The classification of candidates present in multiple major clades reflects the outcome of this reanalysis, not an automatic exclusion rule applied beforehand.

Following the reviewer's suggestion, we have now made explicit in the revised manuscript:

1. Which taxonomic groups were detected for all candidates. How many major clades originally were detected and how many were detected after reevaluation for each candidate (Supplementary Table 3) (Line 207-208).
2. Whether the newly detected clades are phylogenetically congruent (e.g., closely related lineages) or instead involve phylogenetically distant clades, which complicates an LGT interpretation (142-149).

These results are now explicitly summarized in the revised Results and Supplementary Tables, allowing readers to distinguish between cases that remain compatible with HGT and those better explained by alternative evolutionary scenarios.

Finally, we have amended all instances in the manuscript that claimed strict adherence to Katz (2015) criteria, and we now clearly state that our study represents a systematic re-evaluation under updated genomic sampling, rather than a direct replication of Katz's original filtering framework.

We thank the reviewer again for highlighting this key issue, which substantially improved the clarity, transparency, and robustness of our study.

Other concerns:

L114 - which criteria did the authors use to classify an HGT event as "inconclusive"? Clarify.

In this study, candidates were classified as inconclusive when their phylogenetic trees did not allow a confident distinction between horizontal gene transfer and alternative evolutionary explanations. Specifically, this category includes cases in which homologs were detected in multiple eukaryotic major clades, but the query sequence failed to form a well-supported monophyletic group with sequences from any single major clade. Instead, homologs were distributed across disparate positions in the tree, resulting in a patchy phylogenetic pattern.

Such topologies are consistent with multiple evolutionary scenarios—including differential gene loss, incomplete lineage sorting, and horizontal gene transfer—and therefore do not provide sufficient evidence to either support or refute an HGT hypothesis. Accordingly, these candidates were assigned to the *patchy phylogeny* pattern and classified as *inconclusive*. In the current version, we additionally introduce a new pattern within the inconclusive category, termed "*No eukaryotic homologs*". This pattern includes candidates for which no identifiable eukaryotic homolog could be recovered, based on the query sequence identifiers reported by Katz.

We have now clarified this definition in Results sections of the revised manuscript to make the classification criteria explicit (Lines 142-155).

L119, L270 – The authors rely on a fixed number of top hits to decide if there are sufficient (how many?) prokaryotic hits for any given query sequence. Given that databases have grown in size over the past decade, this is surely going to bias the results in unpredictable ways, especially if the authors consider only 200 top hits (L270). It would have been more robust to filter the hit list based on alignment scores (bitscores) or E-values rather than the sheer number of hits. This is a major source of concern because 192 out of 683 candidate LGTs could not be assessed because of this, according to Table 2.

To address this concern, we revised our homology-search strategy and no longer rely on a single fixed-number cutoff across all taxa. Instead, we implemented two independent, taxon-specific homology searches, one restricted to Eukaryota and another restricted to Prokaryota using -taxonlist argument. This approach ensures that prokaryotic homologs are explicitly and systematically retrieved, regardless of database size or the relative abundance of eukaryotic sequences, thereby mitigating the bias highlighted by the reviewer.

Importantly, this revised strategy required repeating all downstream analyses, including homolog retrieval, multiple sequence alignment, phylogenetic reconstruction, and manual tree evaluation. As a result, candidates previously categorized as unassessable due to insufficient prokaryotic representation under the fixed-hit strategy were re-evaluated under a taxon-balanced framework.

We have now clarified this revised methodology (Lines 416-426 & 435-441) in the Methods section and emphasize that our conclusions are based on these taxon-specific, updated searches, which are more robust to database growth and compositional bias.

L189 – The conclusions from these two papers should be explained succinctly.

The Conclusions section has been revised to more clearly and succinctly contrast the main conclusions of Katz (2015) with those of the present reanalysis, emphasizing how expanded taxon sampling and updated homology searches affect the interpretation of interdomain HGT candidates. These revisions are provided in lines 564-594 of the manuscript.

L220 – Note that the statement in Katz 2015 was motivated by the fact that, back then, the only available excavates were highly reduced parasitic genomes, and only one rhizarian was sampled. These extreme situations in sampling bias are not accounted for in your analysis of database effect on LGT detection.

We believe there may be a misunderstanding regarding this point. Our analysis does not seek to explicitly re-model individual historical cases of extreme sampling bias, but instead aims to evaluate the broader impact of expanded taxon sampling across eukaryotes. With respect to Rhizaria, Katz (2015) reports the inclusion of 22 taxa (see Table 1 in the original article: <https://doi.org/10.1098/rstb.2014.0324>), indicating that the limitation at the time was not the presence of a single representative, but rather restricted phylogenetic breadth and limited genome availability. The subsequent expansion of eukaryotic genomes in current databases further supports our conclusion that apparent clade restriction and inferred HGT signals are highly sensitive to database completeness, including in lineages that were not represented by a single genome in earlier analyses.

L247 – Here and elsewhere in the paper, the authors state that relying solely on gene trees for LGT analysis could be "premature" or otherwise misguided. This discussion section is a good place to put forward alternative methods.

General comment on the discussion – a key finding by Katz 2015 is that most of the interdomain LGT events there described were relatively recent (restricted to one major eukaryotic clade). It would be worth discussing how many of these are now inferred to be older.

We agree that statements cautioning against reliance on single-gene phylogenies should be accompanied by a discussion of complementary approaches, and that the temporal interpretation of the HGT events reported in the original article deserves explicit reassessment (Katz, 2015).

In the revised Discussion, this point has been addressed in two ways. First, we now explicitly acknowledge the limitations of single-gene trees for inferring interdomain LGT, particularly for deep evolutionary events, and emphasize the need for integrative evidence. Specifically, we discuss the value of combining phylogenetic signal with genomic context (e.g., flanking gene composition), taxon-specific homology searches, contamination screening, and comparative analyses across closely related species (Lines 339-348). This framework is now reflected both in the revised Discussion, where we state that anomalous gene tree topologies alone should not be interpreted as definitive evidence of LGT, especially in the presence of alternative explanations

such as differential gene loss, incomplete lineage sorting, or unresolved deep branching relationships.

Second, regarding the age of inferred HGT events, Katz (2015) proposed that most interdomain transfers into eukaryotes are relatively recent, as they are restricted to a single major eukaryotic clade. Our reanalysis largely supports this conclusion for the subset of candidates that retain robust phylogenetic support for HGT: the majority of these remain confined to a single major clade even under expanded taxon sampling (Line 308-311). However, the revised analysis also reveals that a substantial fraction of originally single-clade candidates acquire homologs in additional major clades when databases are updated, often resulting in complex or patchy phylogenetic patterns rather than clear support for ancient horizontal transfer (Line 311-318).

Importantly, these cases do not necessarily imply older HGT events. Instead, the revised Discussion emphasizes that such patterns are frequently ambiguous and may reflect alternative evolutionary scenarios, including vertical inheritance followed by lineage-specific losses or incomplete sampling in earlier datasets. As a result, while a subset of candidates now displays broader taxonomic distributions than originally reported, the data do not provide unambiguous support for widespread ancient interdomain HGT. Rather, the reanalysis reinforces the conclusion that confidently inferring the timing of HGT—particularly for events potentially associated with early eukaryotic evolution—remains challenging (Line 319-330).

We believe that the revised Discussion now more clearly articulates both the methodological limitations of gene-tree-based inference and the extent to which Katz's original conclusion regarding the relative recency of most interdomain HGT events is upheld, refined, or rendered ambiguous under expanded taxon sampling.

L281 - Please provide more details on how AVP was used for this analysis. Related to this: did the authors use IQTREE (L282) or Phyml (L301) for their gene tree inferences? Please clarify.

Specifically, IQ-TREE is executed automatically within the AVP framework as part of its internal workflow for preliminary phylogenetic reconstruction and candidate evaluation. In contrast, PhyML was used in our custom downstream pipeline to infer the final gene trees used for manual inspection, classification, and reporting.

This AVP tool is now explicitly clarified in the revised Methods section (Lines 475-503), where the role of AVP and the use of IQ-TREE are described in detail.

Comments on figures:

Fig. 1 – The legibility of this figure would be much improved if the authors expanded their colour-coding to distinguish between prokaryotic and eukaryotic genes more generally.

We thank the reviewer for this helpful comment aimed at improving the clarity of the manuscript. In the revised version, we have redesigned the figure to include representative examples of the patterns described in Table 2, while also directly comparing the phylogenies originally published in Katz (2015) with those reconstructed in this study. Following the reviewer's suggestion, we now distinguish prokaryotic and eukaryotic labels using different colors. We hope that these changes enhance the clarity of the figure and facilitate interpretation of the results for readers.

References

- Aguirre-Carvajal, K., & Armijos-Jaramillo, V. (2025). Reassessing Interkingdom Horizontal Gene Transfer Suggests Limited Influence on Plant Genomes. *Ecology and Evolution*, 15(12), e72653. <https://doi.org/10.1002/ece3.72653>
- Katz, L. A. (2015). Recent events dominate interdomain lateral gene transfers between prokaryotes and eukaryotes and, with the exception of endosymbiotic gene transfers, few ancient transfer events persist. *Philosophical Transactions of the Royal Society B: Biological Sciences*, 370(1678), 20140324. <https://doi.org/10.1098/rstb.2014.0324>

Second decision letter

MS ID#: bio.062387R1

MS Title: What Impact Do New Homologs Have on Detecting Interdomain Horizontal Gene Transfer in Eukaryotes? A Reassessment of Katz (2015)

Authors: Kevin Aguirre-Carvajal; Vinicio Armijos-Jaramillo

I have now reached a decision on the above manuscript.

The reviewer reports are shown at the bottom of this email.

As you will see, the reviewers raised a number of substantial criticisms that prevent me from accepting the paper at this stage. Even if both reviewers acknowledged a substantial effort to answer their comments, and the analysis of the data has been greatly improved, the main conclusions of the study are still not fully supported by the data. In particular, it should be explicit that this analysis is not a direct reassessment of Katz 2015 since the methodology is very different, and the reasons for labeling events as no-HGT should be better justified (a concern raised by both reviewers). If the manuscript is not significantly rewritten following these guidelines, I am afraid it will not meet the standards for publishing in Biology Open and there will be no further revision of the work.

They suggest, however, that a revised version might prove acceptable, if you can address their concerns. If you think that you can deal satisfactorily with the criticisms on revision, I would be pleased to see a revised manuscript. We would then return it to the reviewers.

At this stage, we also ask you to ensure your manuscript complies with our formatting guidelines. Provided you are able to fully address the referees' comments, we are positive about publication of your paper (we accept over 95% of revision submissions) and therefore hope you won't mind any extra work involved in reformatting your manuscript at this point.

Please upload both a 'clean' version of your Word file, along with a highlighted version clearly showing where you have made changes in the revised manuscript. Please avoid using 'Track changes' in Word files as these are lost in PDF conversion.

I should be grateful if you would also provide a point-by-point response detailing how you have dealt with the points raised by the reviewers in the 'Response to Reviewers' box. Please attend to all of the reviewers' comments. If you do not agree with any of their criticisms or suggestions please explain clearly why this is so.

Reviewer 1

Comments for the author

First, I want to acknowledge that this extensive revision and the authors response address all the methodological concerns I had with the original phylogenetic and homolog detection analyses.

Unfortunately, I still have concerns, and these concerns are a rehash of the original reviews (to a large part, the criticism of Reviewer 2). Specifically, I answered "no" to two criteria in this re-review: (1) Does the manuscript title & abstract accurately reflect the contents of the manuscript, without hyperbole? And (2) Are the manuscript's conclusions supported by the data?

(1) The abstract, and much of the text, makes it very clear that the authors still consider this work to be a direct comparison of Katz (2015). While the authors removed some inaccurate text relevant

to this point (e.g., "we adhered to the criteria established in the original study to distinguish valid from invalid candidates") the authors still state that this study is "[a] comprehensive reassessment [that enables] a direct comparison between the original presence-absence patterns and those obtained under current database coverage" (lines 262-264). Yet, as admitted by the authors in the response, they do not use the same methodology as Katz (2015). The criteria for identifying HGTs are different. Thus, this is not a direct reassessment based on the expanded availability of sequences in public databases, but is instead an entirely new analysis of possible HGT events that Katz identified in 2015.

(2) The conclusions still seem to be based on circular reasoning/begging the question - a problem outlined in the original reviews. In short, the authors seem to have predicated their analysis on the assumption that HGTs are rare and thus set a different and more stringent bar for identifying HGTs than the original study, leading them to observe fewer HGTs and thus conclude that HGTs are rare. Further, the authors have categorized several genes that possibly underwent HGT as "No HGT," leading to a seemingly massive overstatement of the difference between this and Katz 2015. Details below:

* To reiterate: Katz (2015) stated: "1138 genes ... meet our criteria of possible interdomain LGTs." These criteria including being found in "three or fewer major clades of eukaryotes".

* The authors repeatedly make it clear that they only consider 344 of these to be credible lateral/horizontal gene transfers (in their terms, "iHGTs"). Yet, among those they label "No HGT":

- 94 were categorized as "endosymbiotic gene transfers" on the basis that homologs were only found in cyanobacteria. While this is a plausible hypothesis, the authors have not convincingly argued that one can necessarily distinguish between EGT and iHGT, at least not with the information we've been given here.

- 178 have "limited donor evidence," because "the inferred direction of transfer was more consistent with a eukaryote-to-prokaryote scenario." While it is true that this would be different from the prokaryote-to-eukaryote scenario proposed by Katz, it is completely inaccurate and misleading to label these as "No HGT."

- 214 candidates were excluded because they formed "phylogenetic trees in which the query sequence formed a monophyletic group with sequences belonging to multiple eukaryotic major clades, rather than being restricted to a single clade." This has several issues: (A) The authors justify this by stating that "an iHGT hypothesis would require invoking an ancient transfer from bacteria into an early eukaryotic ancestor accompanied by widespread secondary losses." While I am not under the impression that this is necessarily a more likely scenario, it is not clearly far-fetched or impossible, and the authors make no convincing argument as to why we should so strongly this argument from parsimony, thus leading to the label "No HGT." Why are these "No HGT" instead of "inconclusive"? (B) As pointed out in the original reviews, this still seems to simply represent a difference in how Katz and the present authors define horizontal gene transfer. Katz considered any gene that occurred in 3 or fewer major clades as a possible HGT event. (C) Is this not in direct contradiction to the authors response to review? The response states: "We emphasize that candidates were not excluded simply because they involved multiple eukaryotic MCs. Rather, such candidates were classified as inconclusive when phylogenetic topologies were compatible with multiple evolutionary scenarios (e.g., early acquisition followed by differential loss, incomplete lineage sorting, or repeated transfers) and therefore lacked a robust, unambiguous HGT signal." If this is not a contradiction, it is not clear what the authors selection criteria were.

Conclusion: I believe these concerns can assuaged by text edits: (i) making it clear that this is not a direct comparison to Katz 2015, but that it is a new analysis starting with the original candidates from Katz 2015; (ii) reassessing/arguing why the categories mentioned above should be "No HGT" rather than, at the very least, "inconclusive"; and (iii) making it clear in the abstract, and throughout, that the "30%" of HGTs represent those that the authors consider to have "unambiguous support" (a phrase the authors use throughout), which is very different than stating that "30% of the original candidates remain congruent with iHGT" (current abstract).

Reviewer 2

Comments for the author

I am generally satisfied with the authors' response to my queries. I commend their thorough effort to reanalyse their data to address my concerns, and I am particularly satisfied with the updated methods section.

My main concern at this point is that the discussion (L269-272) frames genes with inconclusive phylogenetic distributions in distantly related major clades ("patchy distributions") as following an "alternative phylogenetic pattern" different from interdomain HGTs, when, strictly speaking, they could fit either scenario (unlike EGTs or contamination). Therefore, patchily distributed genes should be listed separately (ie as lacking an explanation, rather than providing an alternative one). Likewise, the authors should consider excluding patchily distributed genes from the denominator when calculating the % of cases that remain iHGTs, where appropriate.

Reviewer's Responses to Questions

Experimental quality

Does each figure have the proper controls?

If 'No', please indicate reasons in Comments for Author box below.

Reviewer #1:

- Yes

Reviewer #2:

- Yes

Were the data analyzed using appropriate statistical tests?

If 'No', please indicate reasons in Comments for Author box below.

Reviewer #1:

- Yes

Reviewer #2:

- Yes

Reproducibility

Were experiments performed using adequate number of biological replicates?

If 'No', please indicate reasons in Comments for Author box below.

Reviewer #1:

- Yes

Reviewer #2:

- Yes

Does the methods section provide sufficient detail to permit reproducibility?

If 'No', please indicate reasons in Comments for Author box below.

Reviewer #1:

- Yes

Reviewer #2:

- Yes

Completeness

Are the manuscript's conclusions supported by the data?

If 'No', please indicate reasons in Comments for Author box below.

Reviewer #1:

- No

Reviewer #2:

- Yes
-

Scholarship

Do the authors cite and discuss the merits of data that would argue for and against their conclusion?

If 'No', please indicate reasons in Comments for Author box below.

Reviewer #1:

- Yes

Reviewer #2:

- Yes
-

Does the manuscript title & abstract accurately reflect the contents of the manuscript, without hyperbole?

If 'No', please indicate reasons in Comments for Author box below.

Reviewer #1:

- No

Reviewer #2:

- Yes
-

Second revision

Author response to reviewers' comments

We thank the reviewer for the valuable comments and constructive feedback. Below, we provide detailed responses to each issue raised in this revision.

Reviewer 1: First, I want to acknowledge that this extensive revision and the authors response address all the methodological concerns I had with the original phylogenetic and homolog detection analyses.

Unfortunately, I still have concerns, and these concerns are a rehash of the original reviews (to a large part, the criticism of Reviewer 2). Specifically, I answered "no" to two criteria in this re-review: (1) Does the manuscript title & abstract accurately reflect the contents of the manuscript, without hyperbole? And (2) Are the manuscript's conclusions supported by the data?

(1) The abstract, and much of the text, makes it very clear that the authors still consider this work to be a direct comparison of Katz (2015). While the authors removed some inaccurate text relevant to this point (e.g., "we adhered to the criteria established in the original study to distinguish valid from invalid candidates") the authors still state that this study is "[a] comprehensive reassessment [that enables] a direct comparison between the original presence-absence patterns and those obtained under current database coverage" (lines 262-264). Yet, as admitted by the authors in the response, they do not use the same methodology as Katz (2015). The criteria for identifying HGTs are different. Thus, this is not a direct reassessment based on the expanded availability of

sequences in public databases, but is instead an entirely new analysis of possible HGT events that Katz identified in 2015.

Answer

We thank the reviewer for this sharp and valuable observation. We agree that our methodology does not exactly replicate that of Katz (2015). The principal differences lie in the selection of comparison organisms and in how orthologous groups are defined. In Katz's study, orthogroups were constructed using OrthoMCL from a curated dataset of 487 eukaryotic, 303 bacterial, and 118 archaeal genomes. Duplicates were then filtered, followed by multiple sequence alignment and phylogenetic reconstruction. Although the original manuscript does not describe in detail how the resulting trees were evaluated, we infer that this step was performed manually.

In contrast, our approach does not rely on OrthoMCL or on a predefined subset of genomes. Instead, we used the full set of sequences available in NCBI, inferring homology through similarity searches with DIAMOND and examining these relationships through phylogenetic reconstruction. The downstream filtering steps and phylogenetic analyses are broadly comparable between the two approaches. A key distinction of our framework is that we explicitly evaluate alternative evolutionary explanations for the observed tree topologies, rather than focusing solely on HGT. We consider this emphasis on alternative scenarios to be a central contribution of our manuscript.

Following the reviewer's suggestion, we have revised the manuscript to remove any implication that we replicated Katz's methodology. For example, at the end of the Introduction we now state: *"To evaluate this premise, we examined the 1,138 candidate interdomain horizontal gene transfer (iHGT) events reported by Katz (2015) using updated homology searches, expanded taxon sampling, and revised phylogenetic evaluation criteria. Rather than replicating the original analytical pipeline, our approach applies an alternative and more explicit set of phylogenetic criteria to identify gene candidates with robust support for prokaryote-to-eukaryote transfer."* We also removed language suggesting a direct methodological replication in the Discussion.

We hope these revisions clearly distinguish our approach from that of Katz and eliminate potential sources of confusion.

(2) The conclusions still seem to be based on circular reasoning/begging the question - a problem outlined in the original reviews. In short, the authors seem to have predicated their analysis on the assumption that HGTs are rare and thus set a different and more stringent bar for identifying HGTs than the original study, leading them to observe fewer HGTs and thus conclude that HGTs are rare.

Answer

This is an interesting point. Our conclusions are informed by our working hypothesis that HGT may be less common in eukaryotes than is often assumed. Importantly, this hypothesis did not arise arbitrarily but from previous observations of similar patterns. To evaluate whether those earlier observations were also reflected in Katz's candidates, we reanalyzed the dataset using two independent approaches.

First, we applied the AVP pipeline to detect iHGT candidates. Using AVP's own classification criteria—*independent of our manual assessment*—approximately 40% of the original Katz candidates were identified as HGT cases, while the remainder were classified as NoHGT or complex. Notably, this outcome mirrors the trend observed in our manual analysis, in which roughly 30% of the original candidates were retained as iHGT cases. In all cases, the number of proposed HGT events decreased relative to the original study and never remained unchanged.

A key observation from our analyses is that candidate status often shifts when additional homologs become available for phylogenetic reconstruction. This is not unexpected: the apparent absence of homologs—particularly eukaryotic ones—is frequently what initially supports an HGT interpretation. When additional homologs are incorporated, especially from eukaryotes, the resulting phylogenetic patterns often change. This phenomenon has been reported previously (e.g., Salzberg et al. (2001)) and does not simply reflect our preconceptions.

The broader issue, in our view, is the tendency to interpret unusual or patchy phylogenetic distributions as evidence of iHGT without sufficiently considering alternative explanations, such as incomplete taxonomic sampling in current databases. For this reason, we classified Katz's candidates into multiple groups according to their potential evolutionary origins, explicitly acknowledging the uncertainty in our inferences and considering alternatives beyond iHGT.

Finally, in response to the reviewer's concern: our reasoning is not circular. The reduction in the number of supported iHGT candidates emerges directly from the data and is consistently observed using two independent methodological approaches in this study.

Further, the authors have categorized several genes that possibly underwent HGT as "No HGT," leading to a seemingly massive overstatement of the difference between this and Katz 2015. Details below:

* To reiterate: Katz (2015) stated: "1138 genes ... meet our criteria of possible interdomain LGTs." These criteria including being found in "three or fewer major clades of eukaryotes".

* The authors repeatedly make it clear that they only consider 344 of these to be credible lateral/horizontal gene transfers (in their terms, "iHGTs"). Yet, among those they label "No HGT":

- 94 were categorized as "endosymbiotic gene transfers" on the basis that homologs were only found in cyanobacteria. While this is a plausible hypothesis, the authors have not convincingly argued that one can necessarily distinguish between EGT and iHGT, at least not with the information we've been given here.

Answer

Thank you for raising this point. You are correct that we cannot definitively distinguish between an interkingdom transfer from cyanobacteria and an endosymbiotic gene transfer (EGT). For this reason, in the manuscript we treat these cases as putative EGTs. This interpretation is primarily based on the phylogenetic patterns observed in most of these candidates: sequences from multiple Archaeplastida lineages cluster with cyanobacterial sequences without clear representation from other bacterial groups. Interpreting these cases as direct iHGT from cyanobacteria would require assuming that the transfer occurred in the common ancestor of Archaeplastida and that we are observing only the descendant lineages that retained the gene. However, this timeframe coincides with the primary endosymbiotic event that gave rise to this group. Given the substantial evidence for gene transfer from endosymbionts to host nuclear genomes (Martin et al., 2002; Keeling, 2024), we consider it more conservative to interpret these cases first as potential EGT. The revised manuscript provides a more detailed explanation of the reasoning behind this decision.

- 178 have "limited donor evidence," because "the inferred direction of transfer was more consistent with a eukaryote-to-prokaryote scenario." While it is true that this would be different from the prokaryote-to-eukaryote scenario proposed by Katz, it is completely inaccurate and misleading to label these as "No HGT."

Answer

We agree with this point and have reclassified these candidates as inconclusive to prevent ambiguity.

- 214 candidates were excluded because they formed "phylogenetic trees in which the query sequence formed a monophyletic group with sequences belonging to multiple eukaryotic major clades, rather than being restricted to a single clade." This has several issues: (A) The authors justify this by stating that "an iHGT hypothesis would require invoking an ancient transfer from bacteria into an early eukaryotic ancestor accompanied by widespread secondary losses." While I am not under the impression that this is necessarily a more likely scenario, it is not clearly far-fetched or impossible, and the authors make no convincing argument as to why we should so strongly this argument from parsimony, thus leading to the label "No HGT." Why are these "No HGT" instead of "inconclusive"?

Answer

This is a very interesting point. To avoid confusion, we have decided to remove the label “No HGT” and reclassify these candidates as “inconclusive.”

The parsimony argument underlying our decision is based on a comparison between two evolutionary scenarios. If we consider these scenarios (acknowledging that additional, more complex explanations could exist) to account for the observed phylogenetic patterns, both require invoking widespread gene losses.

In the first scenario—the iHGT hypothesis—an ancient interdomain horizontal gene transfer event would need to be invoked, followed by extensive secondary losses across multiple eukaryotic lineages. In the second scenario—the vertically transmitted hypothesis—the gene would be interpreted as ancestrally present in eukaryotes and subsequently lost in multiple lineages.

We consider the vertically transmitted hypothesis to be more parsimonious because it requires one fewer evolutionary event to explain the same phylogenetic pattern. This rationale is illustrated in the following figure.

Fig. 1. Alternative evolutionary scenarios leading to complex phylogenetic patterns that may be interpreted as HGT-like topologies. Refer to Figure 2 of Aguirre-Carvajal & Armijos-Jaramillo (2025) (<https://onlinelibrary.wiley.com/doi/pdf/10.1002/ece3.72653>).

(B) As pointed out in the original reviews, this still seems to simply represent a difference in how Katz and the present authors define horizontal gene transfer. Katz considered any gene that occurred in 3 or fewer major clades as a possible HGT event.

Answer

The differences we observe are not limited to our criteria; the AVP approach also diverges substantially from the criteria used by Katz. One of our central aims in this manuscript is to highlight that the label of iHGT is often applied to nearly any unusual phylogenetic topology that involves organisms from different biological domains. We believe this tendency deserves more careful reflection.

Through our analysis, we identify recurring patterns in these atypical phylogenies and outline alternative evolutionary explanations beyond iHGT. As illustrated in Figure 1, when several major clades are involved, it becomes particularly difficult to distinguish iHGT from scenarios consistent with vertical transmission. Rather than proposing an additional arbitrary set of rules, our goal is to examine the phylogenetic patterns themselves and reflect on what they can realistically support.

As discussed in the manuscript, different detection methods consistently yield different sets of iHGT candidates. This variability indicates that current approaches are highly sensitive to methodological factors, such as homolog availability. In this sense, the issue extends beyond a simple disagreement in criteria; it reflects a broader structural limitation in iHGT detection and in the level of evidence currently accepted to support putative iHGT events.

(C) Is this not in direct contradiction to the authors response to review? The response states: "We emphasize that candidates were not excluded simply because they involved multiple eukaryotic MCs. Rather, such candidates were classified as inconclusive when phylogenetic topologies were compatible with multiple evolutionary scenarios (e.g., early acquisition followed by differential loss, incomplete lineage sorting, or repeated transfers) and therefore lacked a robust, unambiguous HGT signal." If this is not a contradiction, it is not clear what the authors selection criteria were.

Answer

To avoid any confusion, we have reclassified all these candidates into the inconclusive category. We agree with the reviewer that it is more conservative and appropriate to consider these cases as compatible with multiple evolutionary explanations rather than assigning them a definitive NoHGT label.

Conclusion: I believe these concerns can be assuaged by text edits: (i) making it clear that this is not a direct comparison to Katz 2015, but that it is a new analysis starting with the original candidates from Katz 2015; (ii) reassessing/arguing why the categories mentioned above should be "No HGT" rather than, at the very least, "inconclusive"; and (iii) making it clear in the abstract, and throughout, that the "30%" of HGTs represent those that the authors consider to have "unambiguous support" (a phrase the authors use throughout), which is very different than stating that "30% of the original candidates remain congruent with iHGT" (current abstract).

Answer

(i) We have made every effort to clarify that our analysis does not replicate the methodology used by Katz. However, our work should not be interpreted as a completely new discovery-based analysis, as we did not identify novel candidates. Rather, we re-evaluated the candidates reported by Katz using newly available information, particularly expanded homolog datasets.

(ii) In the revised version, we replaced the broad *No HGT* category with more specific classifications, providing a clearer and more informative reassignment of candidates.

(iii) We appreciate the reviewer's observation regarding our terminology. Upon reflection, we agree that the expression "*unambiguous support*" was too strong and does not accurately reflect the conclusions of our analysis. Our findings highlight the inherent uncertainty in iHGT inference, particularly given the still incomplete sampling of eukaryotic homologs necessary to make definitive claims. For this reason, the phrase "*remain congruent with iHGT*" more accurately captures the level of support observed in our study. The previous wording overstated the certainty of the evidence and has been revised accordingly.

References

- Aguirre-Carvajal, K., and Armijos-Jaramillo, V. (2025). Reassessing Interkingdom Horizontal Gene Transfer Suggests Limited Influence on Plant Genomes. *Ecol Evol* 15, e72653. doi: 10.1002/ece3.72653
- Keeling, P. J. (2024). Horizontal gene transfer in eukaryotes: aligning theory with data. *Nat Rev Genet* 25, 416-430. doi: 10.1038/s41576-023-00688-5
- Martin, W., Rujan, T., Richly, E., Hansen, A., Cornelsen, S., Lins, T., et al. (2002). Evolutionary analysis of Arabidopsis, cyanobacterial, and chloroplast genomes reveals plastid phylogeny and thousands of cyanobacterial genes in the nucleus. *Proc Natl Acad Sci U S A* 99, 12246-12251. doi: 10.1073/pnas.182432999
- Salzberg, S. L., White, O., Peterson, J., and Eisen, J. A. (2001). Microbial genes in the human genome: lateral transfer or gene loss? *Science* 292, 1903-1906. doi: 10.1126/science.1061036

We thank the reviewer for the valuable comments and constructive feedback. Below, we provide detailed responses to each issue raised in this revision.

Reviewer 2: I am generally satisfied with the authors' response to my queries. I commend their thorough effort to reanalyse their data to address my concerns, and I am particularly satisfied with the updated methods section.

My main concern at this point is that the discussion (L269-272) frames genes with inconclusive phylogenetic distributions in distantly related major clades ("patchy distributions") as following an "alternative phylogenetic pattern" different from interdomain HGTs, when, strictly speaking, they could fit either scenario (unlike EGTs or contamination). Therefore, patchily distributed genes should be listed separately (ie as lacking an explanation, rather than providing an alternative one). Likewise, the authors should consider excluding patchily distributed genes from the denominator when calculating the % of cases that remain iHGTs, where appropriate.

Answer

Thank you for this comment and for your careful attention to detail. In the revised version, we have clarified the relevant lines to avoid implying that a patchy phylogeny constitutes an alternative evolutionary explanation. However, we respectfully disagree with excluding these candidates from the calculation of the iHGT percentage. Our objective was to reassess the complete set of candidates originally reported by Katz. Although candidates displaying patchy phylogenies do not allow a definitive evolutionary interpretation, they were classified as iHGT in the original publication.

Therefore, the inability to confidently support these cases as iHGT events in our reanalysis represents an interesting finding of our study. Removing them from the overall percentage would underestimate the extent of revision that results from our reassessment. For this reason, we consider it methodologically appropriate to retain these candidates in the global calculation, as their ambiguous status is relevant to the conclusions we draw.

Third decision letter

MS ID#: bio.062387R2

MS Title: What Impact Do New Homologs Have on Detecting Interdomain Horizontal Gene Transfer in Eukaryotes? A Reassessment of Katz (2015)

Authors: Kevin Aguirre-Carvajal; Vinicio Armijos-Jaramillo

I am happy to tell you that your manuscript has been accepted for publication in Biology Open, pending our standard publication integrity checks. It was accepted on 25th February 2026.